# PIP$_2$ determines length and stability of primary cilia by balancing membrane turnovers

Simon Stilling[1,2], Theodoros Kalliakoudas[3], Hannah Benninghoven-Frey[1], Takanari Inoue [2] & Björn H Falkenburger [1,3,4✉]

Primary cilia are sensory organelles on many postmitotic cells. The ciliary membrane is continuous with the plasma membrane but differs in its phospholipid composition with phosphatidylinositol 4,5-bisposphate (PIP$_2$) being much reduced toward the ciliary tip. In order to determine the functional significance of this difference, we used chemically induced protein dimerization to rapidly synthesize or degrade PIP$_2$ selectively in the ciliary membrane. We observed ciliary fission when PIP$_2$ was synthesized and a growing ciliary length when PIP$_2$ was degraded. Ciliary fission required local actin polymerisation in the cilium, the Rho kinase Rac, aurora kinase A (AurkA) and histone deacetylase 6 (HDAC6). This pathway was previously described for ciliary disassembly before cell cycle re-entry. Activating ciliary receptors in the presence of dominant negative dynamin also increased ciliary PIP$_2$, and the associated vesicle budding required ciliary PIP$_2$. Finally, ciliary shortening resulting from constitutively increased ciliary PIP$_2$ was mediated by the same actin – AurkA – HDAC6 pathway. Taken together, changes in ciliary PIP$_2$ are a unifying point for ciliary membrane stability and turnover. Different stimuli increase ciliary PIP$_2$ to secrete vesicles and reduce ciliary length by a common pathway. The paucity of PIP$_2$ in the distal cilium therefore ensures ciliary stability.

[1] Department of Neurology, RWTH Aachen University, Aachen, Germany. [2] Department of Cell Biology, Center for Cell Dynamics, Johns Hopkins University, Baltimore, MD, USA. [3] Department of Neurology, TU Dresden, Dresden, Germany. [4] JARA-Institute Molecular Neuroscience and Neuroimaging, Forschungszentrum Jülich GmbH and RWTH Aachen University, Aachen, Germany. ✉email: bfalken@ukdd.de

Primary cilia are antenna-like protrusions on many excitatory and non-excitatory cells[1,2]. Primary cilia are considered sensory organelles since they harbour various receptor types[3,4], but many aspects of their functioning are only being discovered. Mutations in genes encoding ciliary proteins cause ciliopathies in humans[5], highlighting the importance of primary cilia during development.

Only quiescent cells are ciliated. Ciliogenesis is initiated at the centriole in G1/G0 phase[6], and ciliary disassembly is required for cell cycle re-entry[7,8]. For delivery of membrane receptors to the cilium, Golgi-derived vesicles fuse with the periciliary membrane[9,10]. Activated receptors are retrogradely transported and endocytosed at the periciliary membrane[11]. When ciliary G-protein coupled receptors (GPCRs) are activated with blocked retrograde transport, cilia secrete GPCRs in vesicles budding from the ciliary tip[12].

In spite of this continuous membrane turnover, ciliary length remains remarkably stable over time. Stimuli that acutely increase ciliary length include exposure to cobalt or lithium, activation of adenylate cyclase, and reduced intracellular calcium[13-15]. An siRNA-based screen for modulators of ciliary length identified genes involved in cell cycle and microtubule regulation, lipid metabolism, actin cytoskeleton and vesicle trafficking[16]. The importance of tubulin modifications, the actin cytoskeleton and endocytosis for ciliary length were confirmed in subsequent studies[17-22]. In addition, there is a tight interaction between the regulation of autophagy and ciliogenesis/ciliary length, involving for instance mTOR, tuberous sclerosis complex proteins and the ciliary regulated transcription factor Gli2[23-25]. Further proteins affecting ciliary length include the Golgi complex protein giantin, which can be compensated by regulator of calcineurin 2[26], and melanin-concentrating hormone receptor 1[27]. For Chlamydomonas, a detailed model of flagellar length regulation was established that includes competition of tubulin between cellular and flagellar binding sites[28] and diffusion of kinesin motors[29]. Still, no universal regulator of the general framework of ciliary length regulation has emerged.

The membrane of primary cilia is continuous with the plasma membrane but shows a distinct distribution of phosphoinositides. Phosphoinositides are signalling lipids[30-32], and cellular membrane compartments differ in their phosphoinositide composition. Phosphoinositol 4,5-bisphosphate ($PIP_2$) is characteristic for the plasma membrane. The amount of $PIP_2$ in the proximal ciliary membrane is as in the plasma membrane, but $PIP_2$ is low in the distal cilium[33-36]. This can be explained by the presence of specific $PIP_2$ degrading enzymes localized in primary cilia. Loss of function mutations in some of these enzymes cause ciliopathies in humans[37-39], indicating that the paucity of $PIP_2$ in the ciliary membrane is functionally relevant.

In this study, we test the hypothesis that ciliary $PIP_2$ is a central regulator of ciliary length in mammalian cell lines. We assume that both the presence of $PIP_2$ in the ciliary base and the absence of $PIP_2$ from the ciliary tip are functionally relevant. In order to test this hypothesis, we used genetically encoded molecular tools to report ciliary $PIP_2$ in individual, living cells and to acutely synthesize or degrade $PIP_2$ in primary cilia, thus circumventing adaptive changes. Since we did not study motile cilia, we refer to primary cilia simply as cilia from here on. In brief, we found that increasing ciliary $PIP_2$ led to shorter cilia whereas $PIP_2$ depletion led to longer cilia. We then determined the molecular pathway mediating these events and applied the obtained insight to a ciliopathy model.

## Results
### Rapidly inducible manipulation of ciliary $PIP_2$ in living cells.
In order to synthesize $PIP_2$ locally in cilia, we used the chemically-induced dimerization (CID) of two protein domains,

FKBP (FK506-binding protein) and FRB (FKBP-rapamycin binding protein domain), by rapamycin (Fig. 1a, b). YFP-tagged FKBP-PIPK has been used in previous studies for acute synthesis of $PIP_2$ at the plasma membrane[40,41]. It consists of the dimerization domain FKBP fused to an engineered PIP kinase that phosphorylates PIP to $PIP_2$ upon addition of rapamycin (Fig. 1a). We expressed YFP-FKBP-PIPK together with a construct consisting of the serotonin receptor $5HT_6$ as a ciliary anchor[8,34,42], mCherry, and the FRB domain ($5HT_6$-mCherry-FRB). Formation of primary cilia in NIH3T3 cells was induced by starvation for 24 h. $5HT_6$-mCherry-FRB was localised selectively in cilia (Fig. 1c). Prior to addition of rapamycin, YFP-FKBP-PIPK was distributed evenly throughout the cell (Fig. 1c—0 h). Upon addition of rapamycin, we observed translocation of YFP-FKBP-PIPK to the cilium (Fig. 1c—1 h). (In this and in all subsequent figures, the red arrowhead marks the location of the recruited CID-tool to the cilium.) Similarly, we used CID to acutely deplete $PIP_2$ from the cilium using recruitment of YFP-FKBP-Inp54p to the cilium (Fig. 1b). Inp54p is an engineered phosphoinositide 5-phosphatase that was previously used to deplete plasma membrane $PIP_2$[40,43]. Again, $5HT_6$-mCherry-FRB was used as ciliary anchor and rapamycin addition led to translocation of YFP-FKBP-Inp54p to the cilium (Fig. 1d).

In order to validate that recruitment of PIPK and Inp54p indeed change $PIP_2$ in cilia, we used a CFP-tagged pleckstrin homology (PH) domain, CFP-PH(PLCδ1), to monitor $PIP_2$ (Supplementary Fig. S1). At rest, PH(PLCδ1) labelled the initial segment of the cilium (Supplementary Fig. S1a—0 min), consistent with earlier reports by us and others[33,34]. (The yellow arrowhead always marks the location of the PH(PLCδ1) probe in the cilium.) When PIPK was recruited to the cilium, the stretch of the cilium labelled by PH(PLCδ1) increased (Supplementary Fig. S1a—60 min, quantified in Fig. S1b). Both the absolute length of the PH(PLCδ1) stretch in μm and the fractional coverage of the cilium by PH(PLCδ1) increased significantly (Supplementary Figs. S1b and S1d). The stretch of the cilium labelled by PH(PLCδ1) did not change when a kinase dead (KD) mutant of PIPK was recruited to the cilium (Supplementary Figs. S1c and S1d). Conversely, when Inp54p was recruited to the cilium (Supplementary Fig. S2), the stretch of the cilium labelled by PH(PLCδ1) decreased (Supplementary Fig. S2a, quantified in S2b and S2c). The decrease was significant when expressed relative to the ciliary length (Fig. S2c) but not when expressed as absolute values in μm (Fig. S2b), possibly due to the small values. We conclude that acute recruitment of PIPK and Inp54p to the cilium alter the phosphoinositide composition reported by PH(PLCδ1).

Recent reports by others have observed a restriction of PH(PLCδ1) to the transition zone of the cilium in *C. elegans*[35]. We therefore sought to clarify in our cells the position of the PH(PLCδ1) signal with respect to the basal body and the transition zone. To this end, we co-expressed YFP-tagged PH(PLCδ1) together with mCherry-tagged CEP290, a marker of the transition zone[44]. CEP290 fluorescence was localised in a punctum at the proximal tip of both the $5HT_6$-CFP marker and the PH(PLCδ1) signal (Supplementary Fig. S3, white arrow). We conclude that different from *C. elegans*, PH(PLCδ1) is not confined to the transition zone in NIH3T3 cells but extends about half way into the cilium.

### $PIP_2$ increase induces ciliary fission.
After confirming that chemically-induced PIPK recruitment to the cilium leads to $PIP_2$ synthesis, we determined the consequences of this manipulation using time-lapse microscopy in single, living NIH3T3 cells. Cilia were visualized by mCherry-tagged $5HT_6$. During the 3 h following PIPK recruitment we observed ciliary swellings (Fig. 1c—

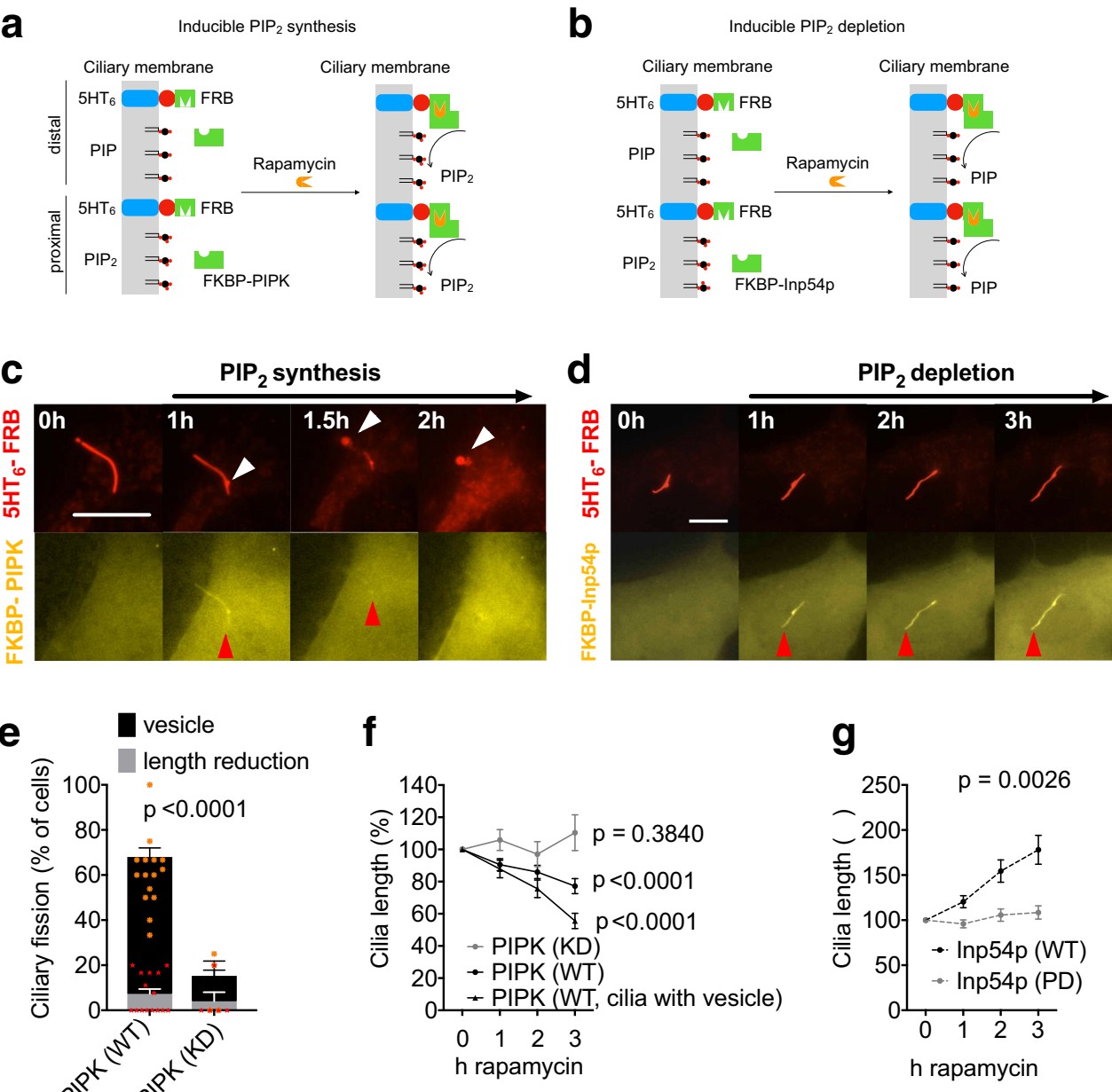

**Fig. 1 PIP$_2$ regulates ciliary length. a**, **b** Schematic representation of inducible PIP$_2$ synthesis (**a**) and PIP$_2$ depletion (**b**). The density of PIP$_2$ is normal in the proximal ciliary membrane but much reduced toward the distal ciliary membrane. Rapamycin-induced dimerization of FRB and FKBP leads to recruitment of the PIP$_2$ synthesising enzyme PIPK (**a**) or the PIP$_2$ depleting enzyme Inp54p (**b**) to the cilium. **c**, **e**, **f** PIP$_2$ increase is sufficient for ciliary fission. **c** Time-lapse images of NIH3T3 cells expressing the ciliary marker 5HT$_6$-mCherry-FRB and the CID tool YFP-FKBP-PIPK (WT) acquired at the indicated time points after addition rapamycin (100 nM final). In all figures, white arrowheads mark ciliary swellings and secreted vesicles. Red arrowheads mark recruitment of PIPK. Scale bar 10 μm. **e** Quantification of ciliary fission events, defined as either appearance of a vesicle at the ciliary tip (black bars) or reduction in ciliary length by at least 20% (grey bars) in cells treated as in (**c**) from $n = 4$–15 independent experiments. Two-way ANOVA showed significant effects of the two factors (PIPK WT vs. DN, and vesicle appearance vs. length reduction) and significant interaction between the two factors. Asterisks represent data of individual experiments, in orange for vesicle appearance and in red for length reduction. **f** Time course of cilium length as reported by mCherry-tagged 5HT$_6$ normalised to the length at $t = 0$. PIPK (WT): $n = 96$ cilia from 15 experiments as in (**c**). PIPK (KD): $n = 18$ cilia from 4 independent experiments as in (**c**) but with kinase dead (KD) PIPK instead of WT PIPK. PIPK (WT, cilia with vesicle): Ciliary length in the subset of cells with WT PIPK where a vesicle appeared. **d**, **g** PIP$_2$ depletion leads to ciliary elongation. **d** Time-lapse images of NIH3T3 cells expressing the ciliary marker 5HT$_6$-mCherry-FRB and in addition the CID tool YFP-FKBP-Inp54p. Images were acquired at the indicated time points after addition of rapamycin (100 nM final). Red arrowheads mark recruitment of YFP-FKBP-Inp54p. Scale bar 10 μm. **g** Time course of cilium length normalised to the length at $t = 0$ h in $n = 58$ cells from 7 independent experiments as in (**d**) and $n = 21$ cells from 3 independent experiments with phosphatase dead (PD) Inp54p instead of WT Inp54p, two-way ANOVA.

1 h, white arrow) and frequently a vesicle dissociating from the distal tip of the cilium (Fig. 1c—1.5 h), leaving behind a shortened cilium (Fig. 1c—2 h). We refer to this process as ciliary fission. (In this and all subsequent figures, white arrowheads mark ciliary swellings and secreted vesicles.) Appearance of a vesicle at the ciliary tip was observed in $60.8 \pm 4.0$ percent of cilia (Fig. 1e, black bars). On average, cilia with such vesicles decreased in length by $44.5\% \pm 4.7$ percent (Fig. 1f, triangles). We hypothesize that not all vesicles remain attached to the cell and that we might miss some fission events if we analyse only the appearance of ciliary vesicles. For all subsequent figures we therefore defined a ciliary fission event as the occurrence of a vesicle at the cilium tip and/or the reduction of ciliary length by at least 20%. A reduction of ciliary length by at least 20% without appearance of a vesicle was observed in $7.3 \pm 2.2$ percent of cells (Fig. 1e, grey bars). Overall, about 68% of cells with recruitment of WT PIPK showed ciliary fission (Fig. 1e, grey and black bars combined), and ciliary length decreased by $22.8 \pm 4.7\%$ when measured across all cells with successful PIPK recruitment (Fig. 1f, black circles). Ciliary fission required the kinase activity of PIPK since fission events were observed much less frequently when a kinase dead (KD) PIPK was recruited (Fig. 1e, $p < 0.0001$, two-way ANOVA), and ciliary length did not decrease with recruitment of KD PIPK (Fig. 1f, grey circles, one-way ANOVA). Vesicle appearance was more common than length reduction (Fig. 1f, $p < 0.0001$, two-way ANOVA), and the difference between WT and KD PIPK was more pronounced for vesicle appearance than for length reduction ($p < 0.0001$, two-way ANOVA).

In order to validate these findings, we used the same CID approach to induce local $PIP_2$ synthesis in the cilium, but in addition to the ciliary marker $5HT_6$-mCherry we stained for acetylated tubulin in order to measure ciliary length (Supplementary Fig. S4a–c). Cells were fixed 3 h after addition of rapamycin. Staining for acetylated tubulin showed a good overlap with labelling by $5HT_6$ (Supplementary Fig. S4a, b). The acetylated tubulin signal after recruitment of WT PIPK was significantly shorter than after recruitment of KD PIPK (Supplementary Fig. S4c). This is consistent with more common ciliary fission for WT as compared to KD PIPK observed with the $5HT_6$ marker for cilia (Fig. 1e) and with the stronger decrease of ciliary length as reported by $5HT_6$-mCherry in live cells (Fig. 1f).

PIPK synthesises $PIP_2$ from PIP. In order to demonstrate that ciliary fission indeed results from the increase in $PIP_2$ and not from the decrease in PIP, we made use of the fact that PH(PLCδ1) domains are not only $PIP_2$ biosensors but also bind and sequester $PIP_2$, thus acting as "$PIP_2$ sponges"[41,45,46]. In an independent series of experiments, we thus compared the consequences of PIPK recruitment with and without co-expression of PH(PLCδ1) (Supplementary Fig. S5a, b). With PH(PLCδ1) co-expression, ciliary fission after PIPK recruitment was observed less frequently (Supplementary Fig. S5c). Taken together, these findings demonstrate that a local increase in ciliary $PIP_2$ leads to ciliary fission.

**$PIP_2$ decrease induces ciliary elongation**. We next determined the consequences of depleting ciliary $PIP_2$. After recruitment of Inp54p to the cilium we observed a continuous elongation of the cilium (Fig. 1d, quantified in 1g). Recruitment of a phosphatase dead (PD) Inp54p mutant did not produce ciliary elongation (Fig. 1g), supporting the notion that elongation is a consequence of the enzymatic activity of Inp54p. The extent of ciliary elongation was substantial, the length of some cilia doubled within 3 h (e.g. Fig. 1d). This elongation was also observed in cells with PH(PLCδ1) (Supplementary Fig. S2), indicating that $PIP_2$ sequestration did not inhibit the effects of $PIP_2$ depletion.

The length of ciliary microtubules is tightly regulated, and changes in ciliary microtubules affect ciliary length[17,20]). We were therefore curious to see whether a manipulation that primarily targets the ciliary membrane can induce changes in the length of ciliary microtubules and used CFP-tagged MAP4m, a life cell marker for ciliary microtubules (Supplementary Fig. S6a). Indeed, we observed an increase in the length of ciliary microtubules as reported by MAP4m. The stretch of MAP4m binding appeared to increase more slowly initially than the length of the ciliary membrane reported by $5HT_6$ (Fig. S6b, 1–3 h), but the difference was not statistically significant (two-way ANOVA), and the length of MAP4m labelling caught up by 24 h after Inp54p recruitment.

Taken together, $PIP_2$ is present in the proximal cilium and absent from the distal cilium (Supplementary Fig. S1a, S2a, S5b), consistent with previous reports[8,34]. An increase in ciliary $PIP_2$ leads to ciliary fission (Fig. 1c, e, f), demonstrating that the absence of $PIP_2$ from the distal cilium is important for ciliary stability. Conversely, a decrease in ciliary $PIP_2$ leads to ciliary elongation (Fig. 1d, g), demonstrating that the presence of $PIP_2$ in the proximal cilium is similarly important. Collectively, these findings suggest that ciliary $PIP_2$ is an important regulator of ciliary length. We next determined the molecular mechanisms by which $PIP_2$ synthesis and $PIP_2$ depletion produce their respective effects.

**Ciliary fission requires actin polymerisation, Rho and Aurora kinases**. The phenotype of ciliary fission was reminiscent of ciliary "decapitation" observed when NIH3T3 or RPE-1 cells disassemble the cilium in order to re-enter the cell cycle after addition of serum[8]. We therefore hypothesized that the molecular mechanism might also be similar and tested the role of actin polymerisation, the Rho family GTPases Cdc42 and Rac1, aurora kinase A (AurkA) and histone deacetylase 6 (HDAC6).

First, PIPK was recruited to the cilium in the presence of the actin depolymerizing agent latrunculin A. With latrunculin A, ciliary fission was observed less frequently than in the control condition, and 3 h after PIPK recruitment cilia remained significantly longer with latrunculin A (Fig. 2a vs. 2b, quantified in 2c, d). In cells expressing dominant-negative (DN) Rac1, we observed less frequent ciliary fission following PIPK recruitment, and ciliary length remained significantly longer (Fig. 2a vs. 2e, quantified in 2g, h). With DN Cdc42 we observed significantly fewer fission events, but the difference in ciliary length was not statistically significant (Fig. 2a vs. 2f, quantified in 2i, j). With latrunculin A, DN Rac1 and DN Cdc42 alone, i.e. without $PIP_2$ synthesis, we observed ciliary fission events at baseline rate; these agents thus did not induce ciliary fission and did not alter ciliary length (Supplementary Fig. S4d–f, quantified in Fig. 2c, d, g–j).

These findings indicate that actin polymerisation and Rac1 are involved in ciliary fission. Yet, latrunculin A and DN Rac1 act upon the entire cell and not only in the cilium. Effects outside the cilium might therefore contribute to their blockade of ciliary fission. To demonstrate that actin polymerisation within the cilium itself is required for ciliary fission, we targeted actin depolymerizing thymosin ß4 (TMSß) to the cilium as described previously[8]. When PIPK was recruited to cilia expressing TMSß, fission was observed less frequently and cilia remained longer 3 h after PIPK recruitment (Fig. 2k vs. 2l, quantified in 2m, n). When TMSß was expressed in cilia but PIPK not recruited, we observed ciliary fission events at baseline rate (Fig. 2m); ciliary length did not decrease but increased (Supplementary Fig. S4g, quantified in Fig. 2n).

To rule out that reduced fission with TMSß results from reduced $PIP_2$ synthesis, we quantified ciliary coverage by

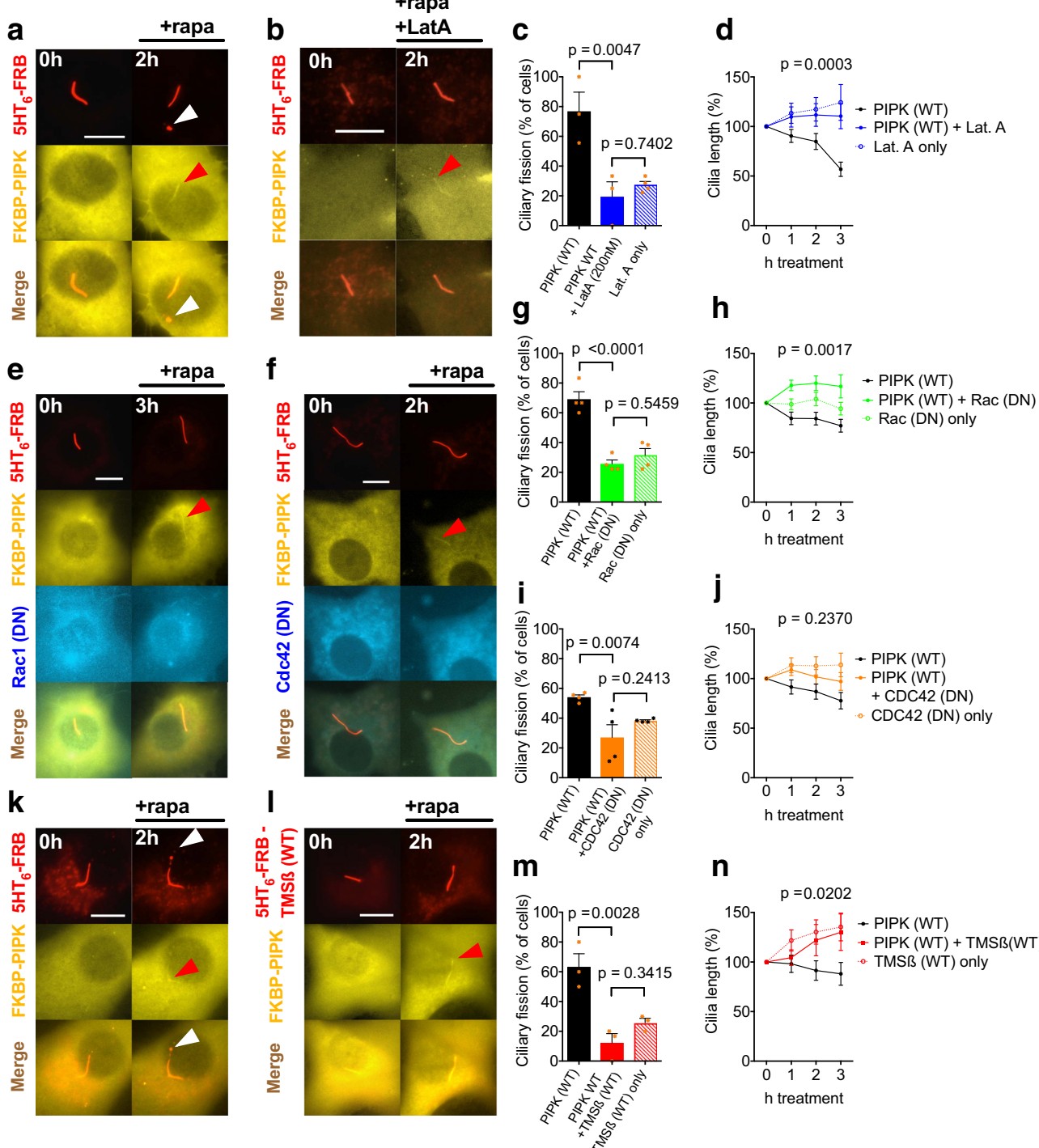

PH(PLCδ1) in cilia with and without TMSß. Ciliary PIP$_2$ as reported by PH(PLCδ1) increased nicely in the presence of TMSß, the extent was comparable to the control condition without TMSß (Supplementary Fig. S7). These findings confirm that ciliary fission requires actin polymerisation in the cilium.

We then tested whether ciliary fission induced by local PIP$_2$ synthesis requires AurkA and HDAC6 using the AurkA inhibitor alisertib and the HDAC inhibitor tubacin (Fig. 3). In the presence of alisertib, ciliary fission occurred less frequently than in the control condition and cilia remained longer 3 h after PIPK recruitment (Fig. 3a vs. 3b, quantified in 3d, e). In the presence of tubacin, ciliary fission occurred less

frequently, but the difference in ciliary length after 3 h was not statistically significant (Fig. 3a vs. 3c, quantified in 3f, g). When cells were incubated with alisertib or tubacin but recruitment was not induced (Supplementary Fig. S4h, i), we observed ciliary fission events at baseline rate (Fig. 3d, f). Ciliary length increased when cells were treated with alisertib or tubacin without PIPK recruitment (Fig. 3e, g).

We conclude that local and acute synthesis of PIP$_2$ in the cilium is sufficient to induce ciliary fission and the signalling cascade involves ciliary actin polymerisation, Rac1 and AurkA. Cdc42 and HDAC6 might contribute to ciliary fission, but the difference in ciliary length was not statistically significant.

**Fig. 2 Ciliary fission requires polymerized actin and Rho GTPase. a**, **b** Time-lapse images of NIH3T3 cells expressing the ciliary marker 5HT$_6$-mCherry-FRB and the CID tool YFP-FKBP-PIPK acquired 0 and 2 h after adding rapamycin (100 nM final) and in addition latrunculin (200 nM final, in **b** only). White arrowheads mark ciliary vesicle. Red arrowheads mark recruitment of PIPK. Scale bar 10 μm. **c** Frequency of ciliary fission in $n = 3-4$ independent experiments with 24 cells treated as in (**a**), 18 cells treated as in (**b**) and 29 cells treated as in Supplementary Fig. S4D. One-way ANOVA: $p = 0.0048$, results of posthoc tests are depicted. **d** Normalized cilium length reported by mCherry-tagged 5HT$_6$ in the same $n = 18$, $n = 24$ and $n = 29$ cells as in (**c**). Two-way ANOVA was used to compare PIPK (WT) and PIPK WT + Lat. A, $p$-value for the interaction between factors treatment and time is depicted. There was no significant difference between PIPK (WT) + Lat. A and Lat. A only ($p = 0.8827$, two-way ANOVA). **e**, **f** Example images of cells as in (**a**), but expressing in addition CFP-tagged DN Rac1 (**e**) or DN Cdc42 (**f**). **g** Frequency of ciliary fission in $n = 4$ independent experiments with 20 cells treated as in (**a**), 32 cells treated as in (**e**) and 40 cells treated as in Supplementary Fig. S4E. One-way ANOVA: $p < 0.0001$, results of posthoc tests are depicted. **h** Normalized cilium length in the same $n = 20$, $n = 32$ and $n = 40$ cells as in (**e**). Two-way ANOVA was used to compare PIPK (WT) and PIPK WT + Rac1 (DN), $p$-value for the interaction between factors treatment and time is depicted. There was no significant difference between PIPK (WT) + Rac1 (DN) and Rac1 (DN) only (two-way ANOVA, $p = 0.0898$). **i** Frequency of ciliary fission in $n = 4$ independent experiments with 33 cells treated as in (**a**), 35 cells treated as in (**f**) and 39 cells as in Supplementary Fig. S4f. One-way ANOVA: $p < 0.0125$, results of posthoc tests are depicted. **j** Normalized cilium length in the same $n = 33$, $n = 35$ and $n = 39$ cells as in (**i**). Two-way ANOVA was used to compare PIPK (WT) and PIPK WT + Cdc42 (DN), $p$-value for the interaction between factors treatment and time is depicted. There was no significant difference between PIPK (WT) + Cdc42 (DN) and Cdc42 (DN) only (two-way ANOVA, $p = 0.4568$). **k**, **l** Time-lapse images of NIH3T3 cells expressing the ciliary marker 5HT$_6$-mCherry-FRB (**k**), respectively, 5HT$_6$-mCherry-FRB-TMSß for actin depolymerisation (**l**) and in addition the CID tool YFP-FKBP-PIPK. Images were acquired 0 and 2 h after addition rapamycin (100 nM final). White arrowheads mark ciliary vesicles. Red arrowheads mark recruitment of PIPK. Scale bars 10 μm. **m** Frequency of ciliary fission in $n = 3$ independent experiments with cells treated as in (**k**), (**l**) and Supplementary Fig. S4G. One-way ANOVA: $p < 0.0028$, results of posthoc tests are depicted. **n** Length of $n = 21$, $n = 18$ and $n = 31$ cilia as in (**m**). Two-way ANOVA was used to compare PIPK (WT) and PIPK WT + TMSß (WT), $p$-value for the interaction between factors treatment and time is depicted. There was no significant difference between PIPK (WT) + TMSß (WT) and TMSß (WT) (two-way ANOVA, $p = 0.8611$).

**Ciliary elongation requires Golgi-derived vesicles**. Membrane proteins enter the cilium by lateral diffusion from the plasma membrane, or by fusion of Golgi-derived vesicles in the periciliary membrane[9,10]. Conversely, membrane proteins leave the cilium by retrograde transport and endocytosis in the periciliary membrane[11]. Since endocytosis requires PIP$_2$[47] whereas exocytosis does not, we hypothesized that ciliary elongation after PIP$_2$ depletion results from an imbalance between membrane removal and insertion. We therefore tested whether ciliary elongation induced by PIP$_2$ depletion can be prevented by interfering with trafficking of Golgi-derived vesicles to the plasma membrane. We first used the antibiotic Brefeldin A (BFA), which inhibits the secretory pathway at the Golgi[48]. Consistent with our hypothesis, cilia 3 h after Inp54p recruitment were shorter with Brefeldin A than without (Supplementary Fig. S8a–c).

In a complementary approach, we inhibited vesicle targeting to the periciliary membrane by dominant-negative (DN) Rab8, a manipulation previously established by others[49]. In cells with WT Rab8, PIP$_2$ depletion caused ciliary elongation whereas ciliary elongation was less pronounced in cells with DN Rab8. Three hours after Inp54p recruitment, cilia were significantly shorter with DN Rab8 than with WT Rab8 (Supplementary Fig. S8d–f). We conclude that ciliary elongation after PIP$_2$ depletion requires insertion of Golgi-derived vesicles into the plasma membrane.

**Ciliary PIP$_2$ mediates vesicle secretion**. Taken together we have demonstrated that ciliary PIP$_2$ is an important regulator of ciliary length. Changes in PIP$_2$ can induce ciliary fission or elongation—by different mechanisms. The molecular mechanism of PIP$_2$-induced ciliary fission shows considerable overlap with ciliary disassembly before cell cycle re-entry. We were therefore interested to see whether further phenotypes are also regulated by the same signalling pathway. Budding of vesicles from the distal tip of primary cilia was recently described when ciliary GPCR are activated but cannot be retrogradely transported and endocytosed for deactivation[12]. This phenomenon was observed in cilia of IMCD3 cells, with receptors for somatostatin (SSTR3) or neuropeptide Y (NPY2R). Agonist-induced vesicle budding is blocked by latrunculin A[12], but the role of ciliary PIP$_2$ has not been investigated.

First, we aimed to reproduce agonist-induced vesicle budding. We therefore applied serotonin to cells expressing fluorescently tagged 5HT$_6$ receptors and a DN dynamin construct to block GPCR endocytosis (Fig. 4a, b). In these cells, serotonin-induced 5HT$_6$-positive puncta at the ciliary tip, likely vesicles (Fig. 4a, white arrow). Appearance of vesicles was significantly more common after addition of serotonin than after addition of buffer only (Fig. 4e, black bars, $p = 0.0033$, two-way ANOVA and posthoc test). As after PIP$_2$ synthesis, we determined the frequency of a reduction in ciliary length by at least 20% (Fig. 4e, grey bars), which occurred less frequently than vesicle appearance ($p = 0.0001$, two-way ANOVA). Overall, ciliary length decreased after serotonin, and 3 h later cilia were significantly shorter with serotonin than with buffer (Fig. 4f, two-way ANOVA). We thus defined a budding event by the same criteria used for ciliary fission, i.e. appearance of a vesicle at the cilium tip and/or the reduction of ciliary length by 20%. Budding events were not observed with either serotonin or buffer when WT dynamin was expressed instead of DN dynamin (Fig. 4g), and ciliary length did not change with WT dynamin (Fig. 4h). These results are consistent with previous findings[12] and indicate that agonist-induced vesicle budding is not only observed with blocked retrograde transport but also with blocked endocytosis. Of note, activation of the GPCR for melanin-concentrating hormone significantly reduced ciliary length in retinal pigment epithelium cells[27]. It is therefore tempting to speculate that activation of some GPCRs might induce ciliary fission even without inhibition of endocytosis or retrograde transport.

We then tested whether actin polymerisation is required for agonist-induced vesicle budding by targeting, as above, the actin depolymerizing protein TMSß to the cilium (Fig. 4c). With this manipulation, fewer budding events were observed upon serotonin addition (Fig. 4i), and cilia were significantly longer 3 h after addition of serotonin (Fig. 4j). These findings confirm that actin polymerisation is required for agonist-induced vesicle budding. In extension of previous work by others[12], who used drugs to block actin polymerisation, our findings demonstrate that actin polymerisation in the cilium itself is required for agonist-induced vesicle budding.

We then tested whether ciliary PIP$_2$ is required for agonist-induced vesicle budding. To this end, we depleted ciliary PIP$_2$ by

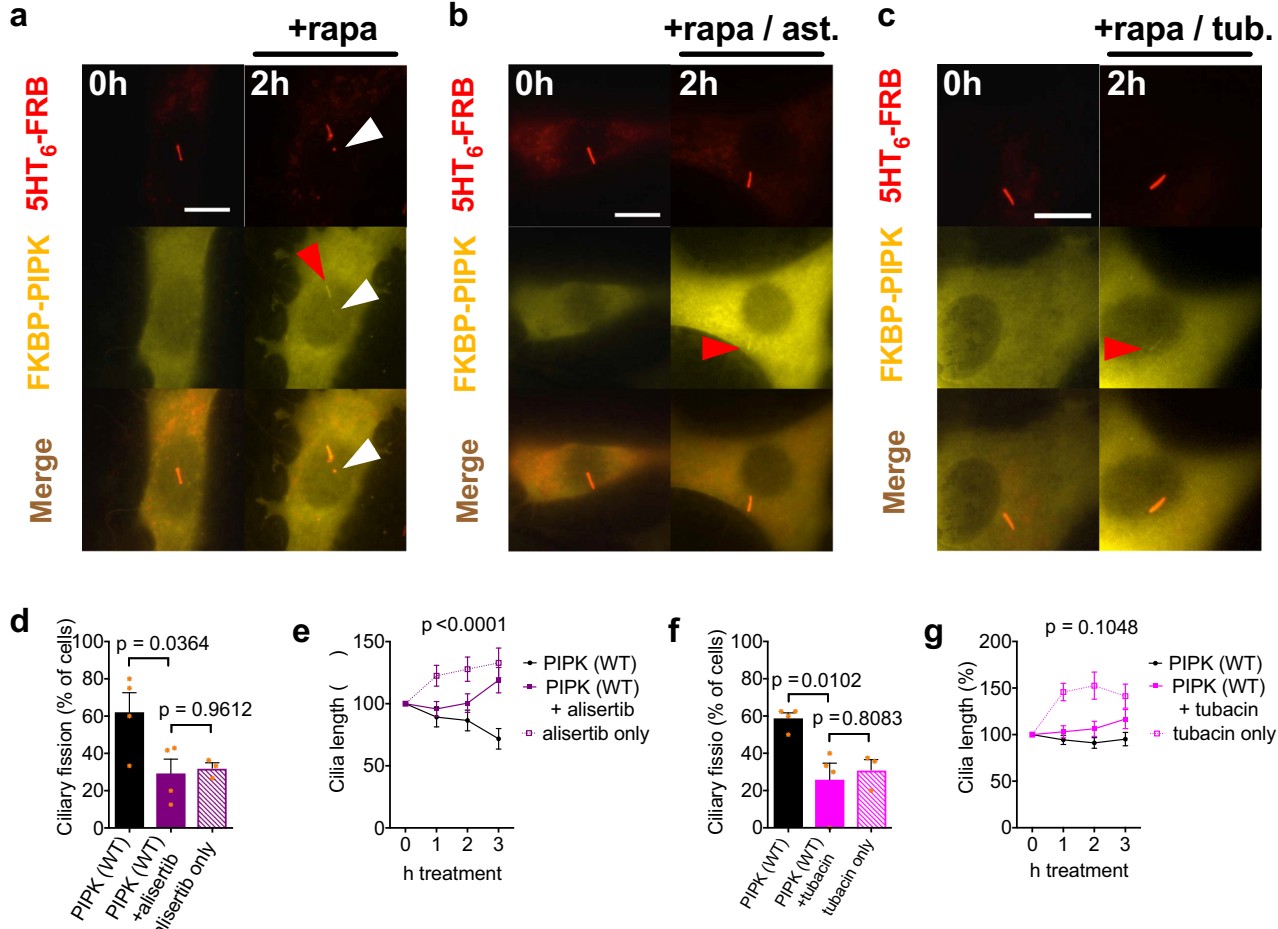

**Fig. 3 Ciliary fission requires Aurora Kinase A. a–c** Time-lapse images of NIH3T3 cells expressing the ciliary marker 5HT$_6$-mCherry-FRB and the CID tool YFP-FKBP-PIPK acquired 0 and 2 h after addition of rapamycin (100 nM final) and in addition alisertib (200 nM final, **b**), tubacin (20 μM final, **c**) or control (DMSO, **a**). White arrowheads mark ciliary vesicles. Red arrowheads mark recruitment of PIPK. Scale bars 10 μm. **d** Frequency of ciliary fission in $n = 3$–4 independent experiments with cells treated as in (**a**), (**b**) and Supplementary Fig. S4h. One-way ANOVA: $p < 0.0402$, results of posthoc tests are depicted. **e** Normalized cilium length in the same $n = 35$, $n = 28$ and $n = 41$ cells as in (**d**). Two-way ANOVA was used to compare PIPK (WT) and PIPK WT + alisertib, $p$-value for the interaction between factors treatment and time is depicted. There was no significant difference between PIPK (WT) + alisertib and alisertib only (two-way ANOVA, $p = 0.0677$). **f** Frequency of ciliary fission in $n = 3$–4 independent experiments with cells treated as in (**a**), (**c**) and Supplementary Fig. S4l. One-way ANOVA: $p < 0.0127$, results of posthoc tests are depicted. **g** Normalized cilium length in the same $n = 38$, $n = 29$ and $n = 40$ cells as in (**f**). Two-way ANOVA was used to compare PIPK (WT) and PIPK WT + tubacin, $p$-value for the interaction between factors treatment and time is depicted. There was also a significant difference between PIPK (WT) + tubacin and tubacin only (two-way ANOVA, $p = 0.0064$).

constitutively fusing the PIP$_2$ phosphatase Inp54p to 5HT$_6$ as previously[8]. When this 5HT$_6$-Inp54p construct was co-expressed with DN dynamin (Fig. 4d), addition of serotonin induced significantly fewer budding events than without Inp54p (Fig. 4k), and cilia remained longer (Fig. 4l). This finding demonstrates that agonist-induced vesicle secretion requires ciliary PIP$_2$ and indicates that it is mediated by the same molecular pathway as PIP$_2$-induced ciliary fission and serum-induced ciliary disassembly.

We then determined whether serotonin binding to 5HT$_6$ receptors changes ciliary PIP$_2$. Ciliary coverage by the PIP$_2$ biosensor PH(PLCδ1) was measured before and after exposure to serotonin or control in the presence of DN dynamin (Fig. 5a, b). Within 1 h of serotonin treatment, coverage of the cilium by PH(PLCδ1) increased from 32.58 +/− 4.93% to 41.79 +/− 4.75% (Fig. 5c), which was significantly more than with vehicle only (Fig. 5c). Together with the observations that PIP$_2$ is necessary for agonist-induced vesicle budding (Fig. 4), and that PIP$_2$ synthesis is sufficient for vesicle secretion (Fig. 1), this finding indicates that local synthesis of PIP$_2$ in the primary cilium mediates budding of ciliary vesicles upon stimulation of ciliary serotonin receptors.

**Blocking the actin/AurkA/HDAC6 pathway reduces PIP$_2$-induced ciliopathy.** Collectively, we have demonstrated that increasing ciliary PIP$_2$ induces ciliary fission (Fig. 1); the same pathway mediates serum-induced ciliary decapitation[8] and agonist-induced vesicle budding (Figs. 4 and 5). Missense mutations in the PIP$_2$ 5-phosphatase Inpp5e cause ciliopathies[38,39] and are expected to increase ciliary PIP$_2$. We therefore hypothesised that this ciliopathy is caused by the same PIP$_2$ dependent pathway as ciliary fission. Accordingly, the swellings we observed after PIP$_2$ synthesis (Fig. 1c—1 h) were comparable to observations by electron microscopy in cilia of renal cells and primary fibroblasts from Inpp5e-/- mice (ref. [39], Fig. 2b; ref. [38], Fig. 4). In order to test our hypothesis, we investigated cells with constitutively increased ciliary PIP$_2$, achieved either by constitutively targeting PIPK to the cilium (Fig. 6a) or by Inpp5e knock-down (Supplementary Fig. S9). Thus, cells were transfected with the ciliary marker 5HT$_6$-YFP and in addition with 5HT$_6$-mCherry-PIPK or 5HT$_6$-mCherry as a control (Fig. 6a, h). Cells were fixed 24 h after transfection. Ciliary length was determined both from the fluorescence of YFP and from staining against acetylated tubulin

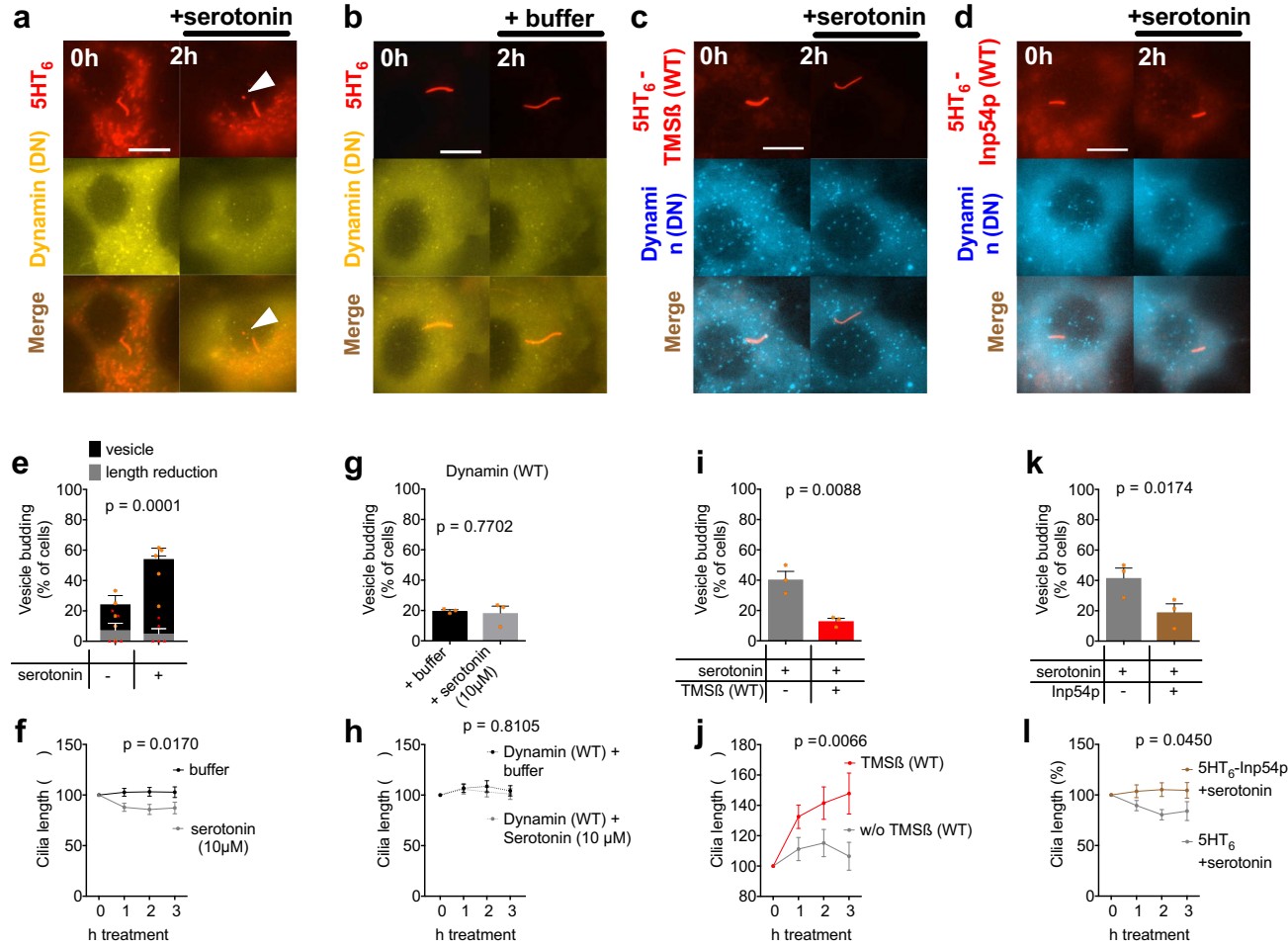

**Fig. 4 Agonist-induced vesicle secretion requires PIP$_2$ and polymerised actin. a, b** Time-lapse images of a NIH3T3 cells expressing the ciliary marker 5HT$_6$-mCherry and in addition GFP-Dynamin2 (DN) to inhibit 5HT$_6$ endocytosis. Images were acquired 0 and 2 h after adding serotonin (10 µM final) (**a**) or buffer (**b**). White arrowheads mark ciliary vesicles. Scale bar 10 µm. **c, d** Images of cells expressing 5HT$_6$-mCherry-TMSß (**c**), respectively, 5HT$_6$-mCherry-Inp54p (WT) (**d**) and in addition CFP-tagged DN Dynamin2. Images were acquired 0 and 2 h after adding serotonin (10 µM final). **e** Frequency of ciliary vesicle budding in $n = 5$ independent experiments with cells treated as in (**a**) and (**b**), two-way ANOVA. Orange asterisks indicate individual data points for appearance of a vesicle, red asterisks indicate individual data points for length reduction. **f** Normalized cilium length in the same $n = 46$ and $n = 61$ cells as in (**e**), p-value is interaction from two-way ANOVA. **g** Frequency of ciliary vesicle budding in $n = 3$ independent experiments with cells treated as in (**a**) but with WT Dynamin instead of DN Dynamin, t-test. **h** Normalized cilium length in the same $n = 45$ and $n = 47$ cells as in (**g**), p-value is interaction from two-way ANOVA. **i** Frequency of ciliary vesicle budding in $n = 3$ independent experiments with cells treated as in (**a**), (**c**), t-test. **j** Normalized cilium length in the same $n = 47$ and $n = 38$ cells as in (**i**), two-way ANOVA. **k** Frequency of ciliary fission in $n = 3$ independent experiments with cells treated as in (**a**), (**d**), t-test. **l** Normalized cilium length in the same $n = 39$ and $n = 37$ cells as in (**k**), two-way ANOVA.

(AcTub). Ciliary length ranged between <2 µm and >20 µm (Fig. 6b, i, l). The median length of cilia was around 8 µm in the control conditions of different experiments. It was similar when using YFP (Fig. 6c, j, m) or acetylated tubulin (Fig. 6d, k, n) as markers. Constitutively targeting PIPK to cilia using 5HT$_6$-mCherry-PIPK reduced ciliary length, as seen by a higher number of short cilia (Fig. 6b) and by a reduced median ciliary length of 4 µm (Fig. 6c, d).

Similarly, cells transfected with Inpp5e siRNA (Supplementary Fig. S9b) showed significantly shorter cilia than cells transfected with control siRNA (Supplementary Fig. S9a), as reported by 5HT$_6$ (Supplementary Fig. S9g) and by acetylated tubulin (Supplementary Fig. S9h). The finding that Inpp5e knockdown reduces ciliary length is consistent with previous work[8,16]. In order to verify that the phenotype of Inpp5e knockdown is indeed mediated by a PIP$_2$ increase, we depleted ciliary PIP$_2$ using 5HT$_6$-Inp54p. This construct localised to cilia (Supplementary Fig. S9c)

and reversed the length difference between Inpp5e knockdown and control cells (Supplementary Fig. S9g, h).

We then tested whether actin polymerisation in the cilium is necessary for the phenotype of increased ciliary PIP$_2$ using 5HT$_6$-TMSß (Fig. 6a and Supplementary Fig. S9d). Targeting TMSß to cilia reversed ciliary shortening observed with expression of PIPK in the cilium (Fig. 6b–d) and with Inpp5e knockdown (Supplementary Fig. S9g, h). We then tested the role of AurkA and HDAC6 for the same phenotypes. The AurkA inhibitor alisertib and the HDAC6 inhibitor tubacin significantly reduced ciliary shortening resulting from expression of PIPK in the cilium (Fig. 6i–n) and also reversed the length difference between Inpp5e knockdown and control cells (Supplementary Fig. S9e, f, quantified in i and j). Of note, AcTub staining was weaker with Tubacin (Fig. 6h), consistent with previous findings by others[50]. Adding the vehicle (DMSO) alone did not alter ciliary length and did not affect the effect of TMSß (Fig. 6e–g). These findings

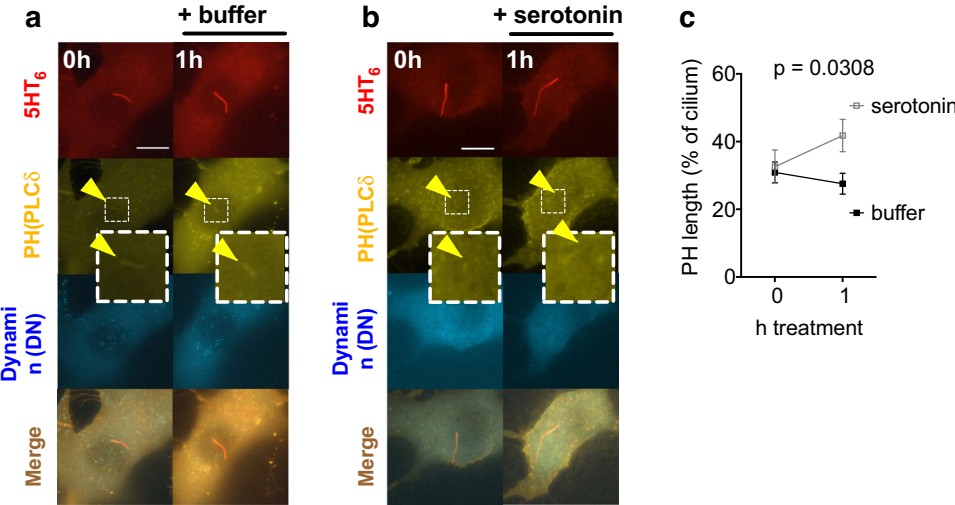

**Fig. 5 Activation of ciliary 5HT$_6$ receptor leads to an increase in ciliary PIP$_2$. a, b** Time-lapse images of NIH3T3 cells expressing the ciliary marker 5HT$_6$-mCherry, CFP-tagged DN Dynamin2 to inhibit 5HT$_6$ endocytosis and the PIP$_2$ biosensor YFP-PH(PLCδ1). Images were acquired before and 1 h after adding buffer (**a**) or serotonin (10 µM final, **b**). Scale bar 10 µm. **c** Length of PIP$_2$ as reported by PH(PLCδ1) expressed relative to the cilium length in $n = 21$ cells treated as in (**a**) and $n = 33$ cells treated as in (**b**), two-way ANOVA.

indicate that ciliary actin, AurkA and HDAC6 are required for the phenotype resulting from increased ciliary PIP$_2$.

## Discussion

In this study we have demonstrated that PIP$_2$ constitutes an important regulator of ciliary length in mammalian cell lines. Increasing ciliary PIP$_2$ was sufficient to induce ciliary fission whereas decreasing ciliary PIP$_2$ led to ciliary elongation. The mechanism of ciliary fission involves actin polymerisation in the cilium, Rho GTPases, AurkA and HDAC6. The same pathway mediates agonist-induced vesicle budding and ciliary shortening resulting from increased ciliary PIP$_2$. Changing ciliary PIP$_2$ is thus engaged by different stimuli as a unifying event to alter ciliary length.

We have mainly exploited the acute recruitment of PIP$_2$ modifying enzymes (PIPK or Inp54p) to primary cilia by CID. While this approach is more complicated than constitutive targeting of enzymes to primary cilia (e.g. using 5HT$_6$-PIPK or 5HT$_6$-Inp54p fusion proteins), acute recruitment avoids compensatory changes that are known to occur with longer-term manipulations. Accordingly, cellular phosphoinositides serve two distinct signalling functions: First, phosphoinositides constitutively identify membrane compartments. For instance, PI(4,5)P$_2$ allows ion channels to function in the plasma membrane, and cytosolic proteins bind to Golgi membranes by recognizing PI4P[51]. Second, brief and/or local changes in phosphoinositide composition are specific cellular signalling events. For instance, several plasma membrane receptors signal through depletion of PI(4,5)P$_2$ by phospholipase C or through synthesis of PI(3,4,5)P$_3$ by PI 3-kinases. Transient local signals include the requirement for specific phosphoinositides during endocytosis[47]. Acute manipulations such as the CID tools used here are better suited to study such transient effects. Yet, our main finding that ciliary fission is dependent on actin, AurkA and HDAC6 was observed both with acute recruitment of PIPK to primary cilia (Figs. 1–3) and with constitutive targeting of PIPK (Fig. 6).

**Regulation of ciliary length by PIP$_2$.** Recruitment of a PIP kinase to the cilium is sufficient to trigger dissociation of a vesicle from the tip of the cilium, leaving behind a shorter cilium (Fig. 1). We refer to this effect as ciliary fission. Cilia and vesicles were

identified by mCherry fused to the ciliary membrane anchor 5HT$_6$, and by staining for acetylated tubulin. Similarly, constitutively targeting a PIP kinase to the cilium reduced median ciliary length (Fig. 6b). We therefore propose that the constitutive absence of PIP$_2$ from the ciliary membrane serves the purpose to prevent ciliary fission and ensure ciliary stability. At the same time, our observation that ciliary PIP$_2$ increases upon exposure to a GPCR agonist (Fig. 5) or serum[8] indicates that different physiological stimuli trigger a rise in ciliary PIP$_2$ in order to induce ciliary length changes. Based on our findings, it is likely that ciliary fission, serum-induced ciliary disassembly and agonist-induced vesicle budding are essentially the same process unified by a PIP$_2$ increase. We still have kept the different terms in order to reflect that these events are triggered by different stimuli and serve different purposes. We argue that this pathway is overactive in Inpp5e dependent ciliopathies. Accordingly, ciliary PIP$_2$ was found increased in Inpp5e-deficient cells[34].

The increase in ciliary PIP$_2$ is not only sufficient to trigger ciliary fission but also necessary: Ciliary fission was not induced by recruitment of KD PIPK (Fig. 1e, f) and reduced by co-expression of PIP$_2$ sequestering PH(PLCδ1) domains (Supplementary Fig. S5). Preventing an increase in ciliary PIP$_2$ blocked serum-induced ciliary decapitation[8] and agonist-induced vesicle budding (Fig. 4i, j). Reducing ciliary PIP$_2$ also ameliorated the ciliary phenotype of Inpp5e knockdown (Supplementary Fig. S9g, h), confirming that it is the rise in PIP$_2$ that is responsible for the ciliary phenotype of Inpp5e knockdown.

Next to the absence of PIP$_2$ from the distal cilium, the presence of PIP$_2$ in the proximal cilium is of similar importance, given that cilia grow if PIP$_2$ is removed from the entire cilium (Fig. 1). We argue that the length of cilia is generally stable because membrane insertion and removal are balanced (Fig. 7a). Cilia thus grow when membrane insertion is increased. In addition, cilia grow when membrane removal is reduced because membrane insertion continues. Endocytosis likely mediates the majority of constitutive membrane removal. It occurs at the periciliary membrane or ciliary pocket[9,10]. Endocytosis requires PIP$_2$[52]. Insertion of Golgi-derived vesicles is described to occur at the periciliary membrane or ciliary pocket[9,10], and there is no indication that it is PIP$_2$ dependent. Fusion of Golgi-derived vesicles is therefore expected to continue after PIP$_2$ depletion. This can explain ciliary growth after PIP$_2$ reduction (Fig. 1). Accordingly, ciliary growth after

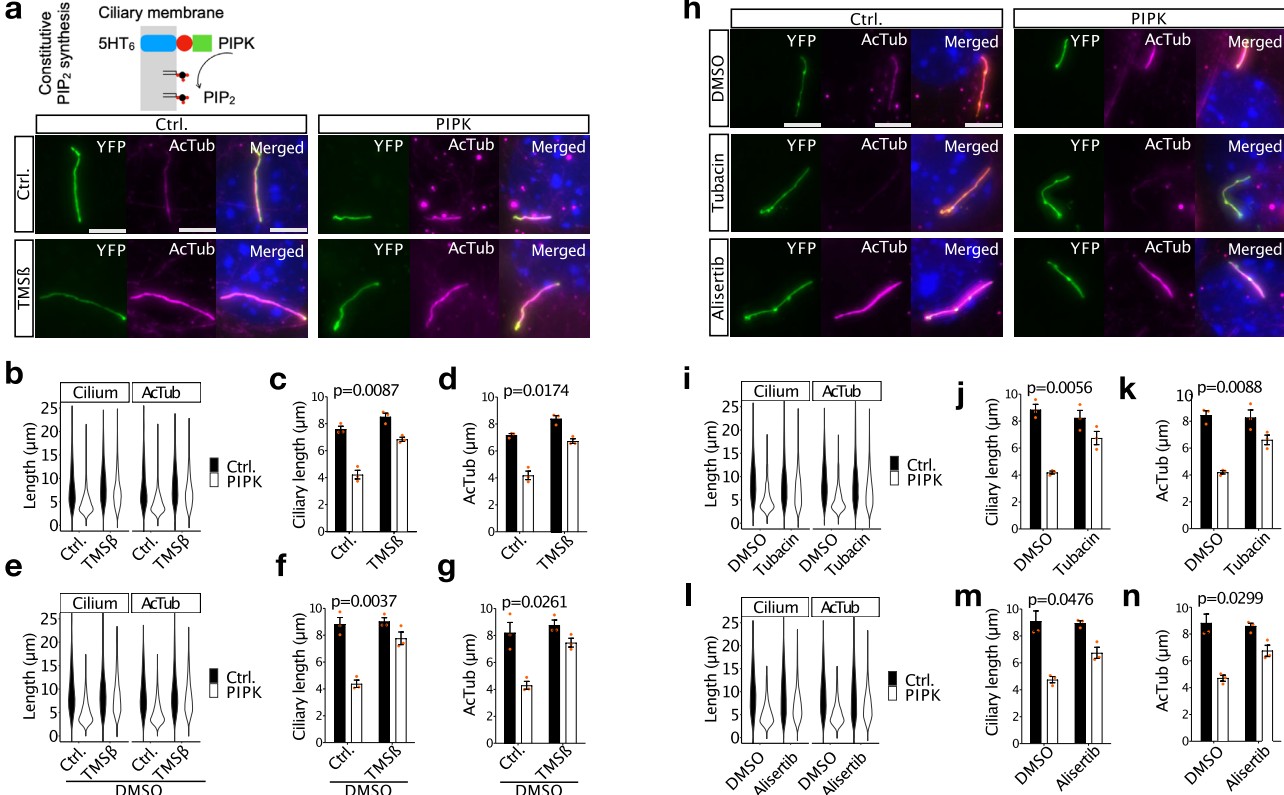

**Fig. 6 Constitutively increasing PIP₂ leads to short cilia. a** Examples of NIH3T3 cells expressing 5HT₆-mCherry-PIPK (right), to constitutively increase ciliary PIP₂, or 5HT₆-mCherry (left), as a control, and in addition 5HT₆-YFP-TMSβ (bottom), to depolymerize ciliary actin, or 5HT₆-YFP (top), as a control. Cells were fixed 48 h after transfection after 18–20 h of starvation. They were stained against acetylated tubulin (AcTub). Scale bars are 5 µm. **b** Violin plot showing all cilia from 3 independent experiments: 241 for Ctrl./Ctrl., 241 for PIPK/Ctrl., 241 for TMSβ/Ctrl., 241 for PIPK/TMSβ. **c, d** Medians of the $n = 3$ independent experiments summarized in (**b**), reporting ciliary length by YFP (**c**) and acetylated tubulin (AcTub, **d**). $p$ values are for the interaction of factors PIPK vs. Ctrl. and TMSβ vs. Ctrl. in two-way ANOVA. **e–g** Quantification of experiments as in (**a–d**), but in the presence of 2 µM DMSO added with the starvation medium (18–20 h before fixation). **e** All cilia: 223 for Ctrl./Ctrl., 223 for PIPK/Ctrl., 223 for TMSβ/Ctrl., 223 for PIPK/TMSβ. **h** Examples of NIH3T3 cells expressing 5HT₆-mCherry-PIPK (right) or 5HT₆-mCherry (left) and in addition 5HT₆-YFP. Cells were exposed to DMSO only (top), 2 µM Tubacin (middle), or 200 nM Alisertib (bottom) added with the starvation medium 18–20 h before fixation. Scale bars are 5 µm. Note that AcTub staining is weaker with tubacin treatment; all images are exposed for the same time. **i** All cilia: 184 for Ctrl./DMSO, 184 for PIPK/DMSO, 184 for Ctrl./Tubacin, 184 for PIPK/Tubacin. **j, k** Medians of the $n = 3$ independent experiments in (**i**). $p$ values are for the interaction of factors PIPK vs. Ctrl. and Tubacin vs. DMSO in two-way ANOVA. **l** All cilia: 175 for Ctrl./DMSO, 175 for PIPK/DMSO, 175 for Ctrl./Alisertib, 175 for PIPK/Alisertib. **m, n** Medians of the $n = 3$ independent experiments in (**l**). $p$ values are for the interaction of factors PIPK vs. Ctrl. and Alisertib vs. DMSO in two-way ANOVA.

PIP₂ depletion was inhibited by brefeldin A and DN Rab8 (Supplementary Fig. S8).

It is tempting to speculate about the possibility that constitutive ciliary fission contributes to membrane removal. The concept of constitutive ciliary fission is supported by the appearance of ciliary vesicles in a small subset of cells even without PIP₂ synthesis (Fig. 1e and Fig. 2c, g, i, m and Fig. 3d, f), in cilia with DN dynamin but without serotonin (Fig. 4e), and in cilia with WT dynamin (Fig. 4g). The role of constitutive ciliary fission could be to offset constitutive vesicle insertion. Consequently, blocking constitutive ciliary fission (by PIP₂ depletion or other means) would lead to ciliary elongation. This hypothesis can explain the spontaneous increase in ciliary length observed without PIPK recruitment when TMSβ was expressed (Fig. 2n) and in cells incubated with alisertib or tubacin but not serotonin (Fig. 3e, g). Constitutive fission is also consistent with our previous finding that ciliary decapitation was observed in about 20% of cells without addition of serum[8]. Finally, the observation by others that blocking actin polymerisation leads to ciliary elongation[16,18] can be explained by the existence of constitutive, actin-dependent ciliary fission.

Of note, vesicle secretion has first been described in non-mammalian cells, including *Chlamydomonas* and *C. elegans* neurons[53–55]. It is not known whether PIP₂ is involved in vesicle secretion in non-mammalian cells. In *C. elegans*, for instance, PI3P and tubulin modifications were found relevant for vesicle secretion[56,57].

We hypothesised that constitutive ciliary fission may offset constitutive vesicle insertion. In order to determine the molecular mechanism by which membrane removal and insertion are balanced, ciliary growth induced by PIP₂ depletion provides an interesting paradigm to investigate the extent of membrane insertion. On average, cilia grew by $78 +/- 16\%$ in 3 h with PIP₂ depletion (Fig. 1g). In some cells, however, cilium length doubled in 3 h. If we use these numbers, at least 20% of ciliary membrane is turned over every hour; an upper limit would be the replacement of most of the ciliary membrane in 3 h.

In this work we have demonstrated that changes originating at the ciliary membrane, PIP₂ synthesis or PIP₂ depletion, can affect ciliary length. This aspect is novel, and further work will be required to resolve how the changes originating at the ciliary membrane are coordinated with changes of the ciliary

## a: Factors affecting ciliary length

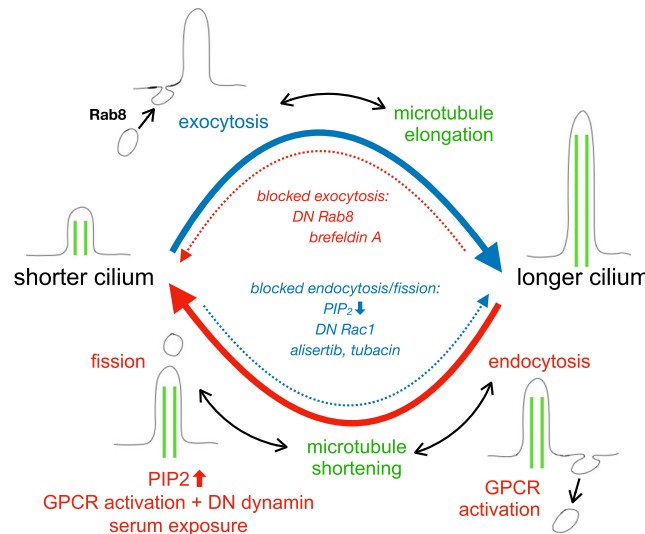

## b: Proposed mechanism of ciliary membrane fission

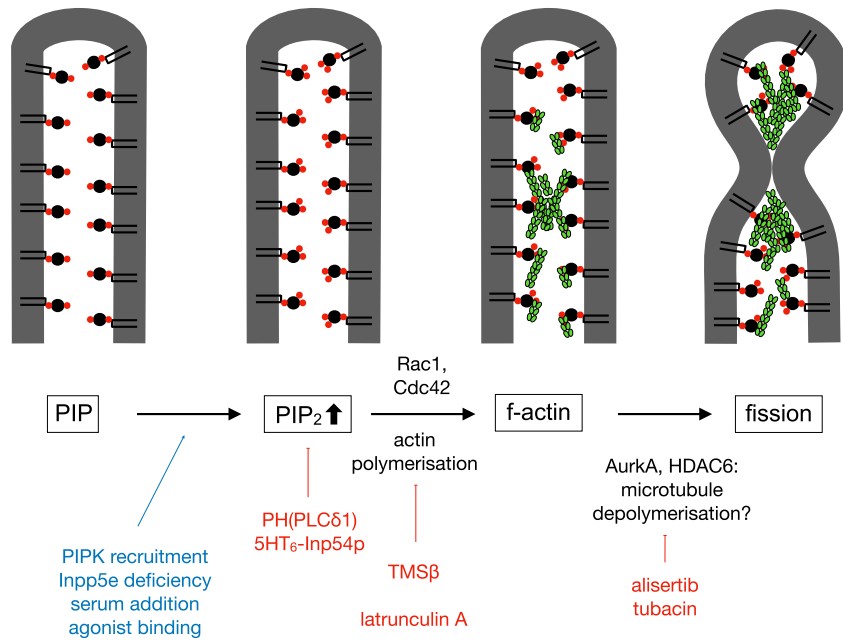

**Fig. 7 Illustration of proposed mechanism. a** Summary of membrane turnover and regulation of ciliary length. Ciliary length remains stable because membrane insertion and membrane removal are balanced. Top: Cilia grow by fusion of Golgi-derived vesicles with the periciliary membrane. Blocking vesicle fusion by DN Rab8 or brefeldin A blocks growth and reduces ciliary length if membrane removal continues. Elongation of the ciliary membrane is accompanied by elongation of ciliary microtubules (green). Bottom: Endocytosis and ciliary fission reduce ciliary length. Fission can be induced by $PIP_2$ increase (synthesis or impaired Inpp5e), activation of G-protein coupled receptors (GPCR) or serum exposure. Blocking these processes by $PIP_2$ depletion, TMSß, DN Rac1, alisertib or tubacin increases ciliary length if vesicle fusion continues. Ciliary microtubules need to be severed for ciliary shortening. **b** Proposed molecular pathway of ciliary fission. Recruitment of PIPK and deficiency of the 5-phosphatase Inpp5e lead to an increase in ciliary $PIP_2$. Serum addition and agonist binding (in the presence of DN dynamin) have the same effect. The $PIP_2$ increase can be prevented by sequestering $PIP_2$ using PH(PLCδ1) or by constitutive expression of the 5-phosphatase Inp54p. The $PIP_2$ increase leads to polymerisation of g-actin to f-actin, presumably amplified by help of the Rho kinases Rac1 and Cdc42. Actin polymerisation can be prevented by TMSß and latruculin A. AurkA and HDAC6 likely act downstream of actin polymerisation and could work by microtubule depolymerisation, which is also required for ciliary fission. AurkA and HDAC are inhibited by alisertib, respectively, tubacin.

cytoskeleton, notably ciliary microtubules. As depicted in Fig. 7a, ciliary microtubules have to be severed for ciliary fission and grow along with the ciliary membrane after PIP$_2$ depletion (Supplementary Fig. S6). Our acute and local manipulation of ciliary PIP$_2$ depletion could represent a useful tool for further studies into how the length of ciliary microtubules is adjusted to the length of the ciliary membrane and vice versa.

**Molecular mechanism of ciliary fission and ciliopathies**. We hypothesise that a similar molecular mechanism (Fig. 7b) mediates ciliary fission, ciliary disassembly in response to serum, vesicle budding induced by receptor activation, and the ciliopathy in Inpp5e-deficient cells.

In this molecular cascade, the first signalling event after PIP$_2$ increase is actin polymerisation. We observed strongly reduced ciliary fission when latrunculin was present (Fig. 2b–d) and when TMSß was targeted to the cilium (Fig. 2l–n). Similarly, decapitation induced by serum was prevented by latrunculin and TMSß[8]. Agonist-induced vesicle budding was blocked by cytochalasinD[12] and by TMSß (Fig. 4i, j). Finally, in cells with constitutive targeting of PIPK to the cilium or Inpp5e deficiency, TMSß rescued ciliary length (Fig. 6b–g, Supplementary Fig. S9g, h). An involvement of the actin cytoskeleton downstream of PIP$_2$ synthesis is plausible because actin-binding proteins use PIP$_2$ to bind to the plasma membrane. From a membrane topology point of view, ciliary vesicle budding shares similarities with cytokinesis, for which an involvement of actin is well established. This conclusion is in line with a substantial body of work indicating that modulators of the actin cytoskeleton affect ciliary length[16,18], including CDC14A[21] and FHDC1[22].

Furthermore, DN Rac1 reduced ciliary fission after PIP$_2$ synthesis (Fig. 2g, h), similar to its effect on serum-induced ciliary decapitation[8]. The effect of DN Cdc42 was less pronounced (Fig. 2i, j). Rac1 and Cdc42 have been implicated in a positive feedback loop for actin polymerisation[58], and small GTPases of the Rho family have been implicated in the regulation of PIP$_2$ synthesis[56,59,60]. The involvement of Rac1 and Cdc42 in ciliary fission is therefore highly plausible, but their exact mechanistic role remains to be determined.

PIP$_2$-induced ciliary fission was inhibited by alisertib (Fig. 3d, e), indicating that AurkA participates in this process. AurkA inhibitors also blocked serum-induced vesicle secretion[8,61] and we have demonstrated that alisertib ameliorates the ciliary phenotype resulting from constitutive PIPK targeting (Fig. 6l–n) or Inpp5e knockdown (Supplementary Fig. S9i, j). These findings indicate that AurkA acts downstream of a rise in ciliary PIP$_2$. Accordingly, AurkA amount and phosphorylation were found increased in Inpp5e-deficient cells[61].

The most plausible mechanistic role for AurkA in ciliary fission is an involvement in microtubule disassembly through HDAC6. Accordingly, the HDAC6 inhibitor tubacin reduced ciliary fission induced by acute PIP$_2$ synthesis (Fig. 3f) and serum-induced ciliary decapitation[8]. Tubacin also ameliorated the ciliary phenotype resulting from constitutive PIPK targeting (Fig. 6i–k) and Inpp5e knockdown (Supplementary Fig. S9i, j). The involvement of HDAC6, which regulates posttranslational tubulin modifications, provides a possible link to the large body of literature demonstrating that posttranslational tubulin modification affects ciliary length, including tubulin glycylation[17] and phosphorylation by tau tubulin kinase 2[20].

AurkA and HDAC6 are good drug targets. Alisertib has been tested in clinical trials against lymphoma[62]. It was well tolerated, so clinical repurposing trials in ciliopathy patients with Inpp5e mutations could be considered.

These findings thus describe a common pathway for different physiological and pathological events. This pathway is triggered by a PIP$_2$ increase. Ciliopathies resulting from Inpp5e mutations can be considered consequences of the permanent activation of this PIP$_2$-dependent ciliary fission pathway.

Ciliopathies can also be caused by mutations in OCRL. OCRL has been involved in diverse cellular events: OCRL is (a) a PIP$_2$ 5-phosphatase[63–65], (b) a GTPase-activation protein (GAP) for Rac1 and Cdc42[66], and (c) important for endocytosis[65,67]. Interestingly, all these effects are involved in the PIP$_2$ dependent fission pathway. Mutations in OCRL could therefore cause its ciliopathy by an increase in PIP$_2$, by overactivity of Rac1 and Cdc42, and by impaired endocytosis (impaired endocytosis may trigger ciliary shortening by causing agonist-induced vesicle budding). The PIP$_2$ dependent ciliary fission pathway we describe here can thus provide a framework for previously unrelated cellular effects.

**Conclusions**. Taken together, we have provided evidence that PIP$_2$ is a unifying determinant of ciliary length. The absence of PIP$_2$ from the distal cilium prevents ciliary disassembly and vesicle secretion. The synthesis of PIP$_2$ in the distal cilium can induce these events, and the presence of PIP$_2$ in the proximal cilium allows membrane turnover. Our findings confirm that ciliopathies can result from an excess of PIP$_2$ and indicate that ciliopathies could be ameliorated by AurkA and HDAC6 inhibitors.

## Methods

**Cell culture and transient transfection**. NIH3T3 mouse embryo fibroblasts were validated by IDEXX GmbH, Ludwigsburg, Germany using STR profiling in January 2018. They were cultured in DMEM Medium (PAN-Biotech, Germany, Aidenbach, or Cat#31966-021 by Thermo Fisher Scientific, Waltham, MA, USA) containing 10% fetal bovine serum (FBS; Biochrom, Germany, Berlin or Anprotec, Bruckberg). Transfection of DNA constructs was done directly after plating using X-treme Gene 9 (Sigma-Aldrich, Cat#6365787001) following the manufacturer's recommendation. Twenty-four hours after transfection, cells were incubated in Opti-MEM (Thermo Fisher Scientific, Cat#31985-062) for 20–24 h to induce quiescence. For live-cell imaging, cells were plated on poly-L-lysine (Sigma-Aldrich, Cat#P4707) coated 24-well plates with glass bottom (Greiner Bio-One, Frickenhausen, Germany) prior to transfection. For imaging fixed cells, cells were plated on poly-L-lysine coated glass cover slips.

**Chemicals**. For rapamycin treatment, a 2.74 mM ready-made solution in DMSO (Sigma-Aldrich, Cat#R8781) was diluted with ddH$_2$O to 100 μM. 0.5 μl was added to cells incubated in 500 μl Opti-MEM to attain 100 nM final concentration. 100 μg Latrunculin A (Sigma-Aldrich, Cat#L5163) was diluted in DMSO (Sigma-Aldrich, Cat#D8418) for a 100 μM stock. This was diluted in ddH$_2$O to 10 μM. 10 μl was added to cells to attain 200 nM final concentration. Tubacin (Cayman Chemical Company, Cat#13691) was dissolved in DMSO for a 2 mM stock. This was diluted with ddH$_2$O to 20 μM and 50 μl was added to cells to attain 2 μM final concentration. Alisertib (MedChem Express Cat#HY-10971) was dissolved in DMSO for a 2 mM stock. This was diluted with ddH$_2$O to 20 μM and 5 μl was added to cells to attain 200 nM final concentration. For these chemicals, an equal concentration of DMSO was used as control. Serotonin (Sigma-Aldrich, Cat#H9523) was freshly dissolved in H$_2$O for a 100 μM stock and diluted to 10 μM with H$_2$O. 50 μl was added to cells to attain 10 μM final concentration. Brefeldin A (Sigma-Aldrich, Cat#B5936) stock (10 mg/ml) was diluted 1:100 with ddH$_2$O for a 100 ng/μl working dilution and 0.5 μl was added to cells to attain 100 ng per 500 μl culturing medium.

*Plasmids*. Fusion of fluorescent proteins to the serotonin receptor 6 (5HT$_6$) was used to label the primary cilium. 5HT$_6$-mCherry, and 5HT$_6$-mcerulean3 (referred to as 5HT$_6$-CFP for simplicity) were described in ref. [8]. Fusion to 5HT$_6$ was also used to constitutively target a phosphoinositide phosphatase (Inp54p) and a phosphoinositide kinase Iγ (PIPK) to the primary cilium. 5HT$_6$-EYFP-Inp54p and the non-functional mutant 5HT$_6$-EYFP-Inp54p(D281A) were described in ref. [34]. 5HT$_6$-mCherry-Inp54p and 5HT$_6$-mCherry-Inp54p (D281A) were created by replacing EYFP by mCherry using AgeI and BsrgI.

For acute recruitment of PIPK (which phosphorylates PIP to PIP$_2$) or Inp54p (which dephosphorylates PIP$_2$ to PIP) to the primary cilium we used rapamycin-induced dimerization of FRB and FKBP. The ciliary anchor 5HT$_6$-CFP-FRB was

reported in ref. [42]. 5HT$_6$-mCherry-FRB was generated by replacing CFP by mCherry using AgeI and BsrgI. The recruitable enzymes CFP-FKBP-PIPK, respectively, CFP-FKBP-Inp54p, and the control plasmids CFP-FKBP, CFP-FKBP-PIPK(D253A), CFP-FKBP-Inp54p(D281A), were described in ref. [40]. YFP-tagged versions were created by replacing CFP by YFP using AgeI and BsrgI.

To suppress intraciliary actin polymerisation we used as previously thymosin ß4 (TMSß)[68]. TMSß was targeted to cilia using fusion to 5HT$_6$ as previously For use with rapamycin-induced dimerization we created 5HT$_6$-mCherry-FRB-TMSß by inserting FRB into 5HT$_6$-YFP-TMSß using AgeI and EcoRI.

As biosensor for PIP$_2$ we used as previously[51] PH(PLCδ1) tagged with CFP or YFP. To label the transition zone of primary cilia we used mCherry fused to Cep290, a gift from Joseph Gleeson (Addgene plasmid #27380). Dominant-negative (DN) versions of the small G-proteins Rac1 and Cdc42 have been used previously[41]. Dynamin (DN) was used to inhibit endocytosis. EGFP-tagged dynamin was a gift from Pietro De Camilli (Addgene plasmids #22197 and #22163 for WT and the DN mutant K44A). GFP was replaced by CFP using AgeI and NotI. To visualise ciliary microtubules we used a CFP-tagged version of the microtubule-associated protein MAP4m[69] as reported previously[70]. To inhibit the transport of vesicles from Trans-Golgi Network to primary Cilia, we used DN Rab8 (T22N). Rab8-T22N(Flag) in pcDNA3.1neo was a gift from Terry Hébert (Addgene plasmid # 46784). Rab8 WT in pcDNA3.1neo was used as control and a gift from Terry Hébert as well (Addgene plasmid # 46783). GFP was replaced by CFP using AgeI and NotI.

*Small interfering RNA (siRNA).* To suppress endogenous Inpp5e, mouse Inpp5e siRNA (Dharmacon, Cat# M-041108-00-0005) was transfected as follows. Inpp5e siRNA was diluted in 1x siRNA Buffer to attain a stock concentration of 20 μM. Transfections were performed with a final siRNA concentration of 50 nM. Cells for siRNA transfection were plated in plastic 6-well plates. The first transfection of siRNA was performed 3–6 h after cell plating using Metafectene (Biontex Laboratories, Munich, Germany) following the manufacturer´s instructions (1 μg siRNA per 2 μl Metafectene). Twenty to twenty-four hours after the first transfection, cells were re-plated on poly-L-lysine coated glass cover slips. Plasmid DNA and a second round of siRNA transfection was performed 24 h after the second plating using Metafectene (1 μg siRNA or DNA per 3 μl Metafectene). Twenty-four hours after transfection, formation of primary cilia was induced by starvation with Opti-MEM for 24 h. Cells were fixed on day 5. Non-targeting siRNA (Dharmacon, Cat# D-001206-13-20) was used as negative control.

*Time-lapse microscopy and quantitative image analyses.* For time-lapse microscopy, we used an Olympus IX81 epifluorescence microscope (×60 oil objective, NA 1.35) equipped with an incubator (37 °C, 5% CO$_2$) and a motor stage to acquire images at defined positions every 30 to 60 min over 3 h. Photomicrographs of living cells were acquired as z-stacks with 10 slices 0.5-μm apart using Olympus imaging software xcellence (2.0). Drugs or vehicle were applied directly into the well after cells were identified and their positions stored.

Recruitment of YFP-FKBP-PIPK or YFP-FKBP-Inp54p into the cilium was achieved by addition of rapamycin (100 nM). Negative controls include the addition of just DMSO instead of rapamycin and the recruitment of non-functional proteins such as just YFP-FKBP or the enzyme-dead variants YFP-FKBP-PIPK (D253A) and YFP-FKBP-Inp54p (D281A). When rapamycin was added, we analysed only cells with successful recruitment, defined as the primary cilium becoming visible in the YFP channel after addition of rapamycin.

Images were analysed offline using ImageJ (NIH) or Olympus imaging software xcellence (2.0). To measure cilium length, a segmented line was traced along the signal of the ciliary marker (5HT$_6$-mCherry or 5HT$_6$-CFP). To measure the extension of PH(PLCδ1) or CFP-MAP4m in the cilium, a segmented line was traced along the signal of PH(PLCδ1) or CFP-MAP4m and expressed (a) as absolute length in μm and (b) relative to the length of the cilium as defined by the ciliary marker. The event "fission" was defined by the appearance of at least one 5HT$_6$-mCherry positive vesicle in the region of the ciliary tip or by a decrease in cilium length by more than 20% of its initial length.

**Fixed cells and immunocytochemistry.** Cells were fixed with 4% PFA in 5% sucrose. Immunocytochemistry was performed with a monoclonal primary antibody against acetylated tubulin (AcTub, Sigma-Aldrich, Cat# T6793 or Cat# T7451). A DyLight 405-labelled goat anti-mouse IgG (ThermoFisher, Cat# 35501BID) or an Alexa Fluor 647-labelled donkey anti-mouse IgG (ThermoFisher, Cat#A31571) were used as secondary antibodies. Images were acquired and analysed as described above for living cells or with a Z.1 Observer microscope (Zeiss, Oberkochen, Germany). Perinuclear structures positive for acetylated tubulin were considered cilia when >0.5 μm in length.

**Statistics and reproducibility.** We used GraphPad Prism 5 or 6 (GraphPad Software, La Jolla, USA) for data illustration and statistical analysis. Graphs represent mean ± SEM. In addition, we included the individual values as asterisks. "*p*" values are noted in the graphs. $p < 0.05$ was considered statistically significant. The number of cells and independent experiments summarized and the test used for comparison are noted in the figure legend for each graph. For the comparison of the frequency of ciliary

fission in two groups as in Fig. 1e, we calculated the % of cells showing ciliary fission for each experiment and compared the percentages between experiments with "*n*" corresponding to the number of experiments. For time series of one parameter but two groups as in Fig. 1g we report the *p*-value of the interaction of the factors time and group from a repeated measures two-way ANOVA. The violin plots in Fig. 6 were generated using Rstudio 1.3 and the ggplot2 package.

**Reporting summary**. Further information on research design is available in the Nature Research Reporting Summary linked to this article.

## Data availability

All data are included in the manuscript text, figures and Supplementary Information. Raw data are in Supplementary Data 1. Plasmids are available from the authors upon request.

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

## Acknowledgements

We thank Jörg Vervoorts-Weber for NIH3T3 cells and the acetylated Tubulin antibody and Theodora Saridaki for help with cloning. We also thank Siew Cheng Phua for helpful discussions and technical advice. This work was supported by the Department of Neurology at RWTH University Aachen, the JARA-Institute Molecular Neuroscience and Neuroimaging and the US National Institutes of Health (grant DK102910 to T.I.).

## Author contributions

T.I. and B.F. conceived research. S.S., T.K. and H.B. performed research. S.S., T.K., H.B. and B.F. analysed data. S.S. and B.F. wrote the first draft of the manuscript. All authors contributed to writing and approved the final version of the manuscript.

## Funding

## Competing interests

The authors declare no competing interests.
