## [Transparent Peer Review File · Communications Biology]

Reviewers' comments:

Reviewer #1 (Remarks to the Author):

In this manuscript, the authors use the FKBP-FRB system to rapidly change the levels of PIP2 in the cilium, by inducing the localization of either a PIP2 kinase or a phosphatase into the cilium. Using live imaging and fixed cells they show that increased levels of PIPK and thus increased levels of PIP2 in cilia results in reduced ciliary length and increased ciliary fission. In contrast, depleting PIP2, by recruiting INP54P to the cilium, resulted in longer cilia. The authors identify several factors that mediate the effects of increased PIP2 on cilium length including actin polymerization, most likely in the cilium.

Next, the authors compare their results to a previously identified pathway that regulates shedding of vesicles containing ligand activated receptors when retrograde transport is blocked. Their results suggest that these are indeed very similar processes. Finally, the authors extrapolate their results to Joubert syndrome, a ciliopathy that can be caused by loss of function of INPP5e.

The manuscript reports very interesting results and in principle contributes an important step forward in our understanding of the regulation of cilium length, and links this to a ciliopathy and cellular signaling. However, some of the data presented in the manuscript in its current form are not sufficiently convincing.

Major points

Although figure 1 nicely shows recruitment of PIPK or INP54P to the cilium, I'm not convinced by the results presented in Sup fig 1 and 2. Especially in Sup Fig 1 I don't see much change in the length of the blue region. The authors have quantified these effects in S1b and S2b. Are these changes significant?

The authors define a "fission" event by the appearance of at least one vesicle in the region of the ciliary tip, but also by a decrease in cilium length by more than 20% of its initial length. Although this has been clearly stated in M&M and results, I think the authors should be more careful. Is a decrease in more than 20% really fission; why not rapid resorption? Would it be possible to show this? In addition, the authors should show what fraction of their data consists of "real" fission events and what fraction are the rapid length reductions.

The authors quantify the length of the cilia in Figure 1F. Please indicate if these changes in length are significant. I think that including the length measurements of the cilia that underwent fission is not useful, as this includes the cilia that were reduced in length by more than 20% .

In Figure 2 important controls are missing. What happens when cells are incubated only with LatA, RAC1(DN), CDC42(DN) and ciliary TMS β ? Similarly, the authors should show the results of incubations with only alisertib or tubacin without addition of rapamycin in Figure 3, and only with BFA in Figure S8. And the same applies to results presented in Figure 4

The authors conclude that the effect of PIP2 on cilium length requires actin polymerization, importantly in the cilium. Very interesting, but to me a bit preliminary. Would it be possible to prove this in an independent way, or e.g. show actin polymerization in the cilium?

On page 19 the authors state that they observed budding events in 48.15 + 15.65% of cells exposed to serotonin but in less than 10% of the cells incubated with buffer. However, these results are not significant, as indicated in Figure 4. This should also be stated in the text.

At the beginning of page 19 the authors conclude that agonist induced vesicle budding (which is not significantly different from non-agonist induced "budding") can be induced by blocking endocytosis.

However, this has not been tested: there are no data of cells treated with vs without dynamin (DN) to show that this affects vesicle budding. In addition, it is important that the authors reproduce the published results, by using latrunculin A, to show if latrunculin A affects vesicle budding to the same extent as DN-dynamin, and also to combine the two to see if they have additive effects.

Page 21, Figure 6G-J. These figures indicate there are significant differences, but it is not clear which results are significantly different, and to what extent. The ANOVA shows there are significant differences, but this should be followed by a post-hoc analysis that reveals the significance of specific comparisons.

Minor points

P4, end of 2nd paragraph: please change GPCR for GPCRs

P4, end of 3rd paragraph: please add "the" between "of" and "general"

P5, line 3. The authors state that in the proximal ciliary membrane the amount of PIP2 is "normal". "normal" is not the right term. The authors should better use "as in the plasma membrane" or something similar. The same applies to line 5 of the abstract.

P7, plasmids, 2nd paragraph. Please correct "PIPK (which phosphorylates PIP2 to PIP)".

P14, line 14, please change "Figure 2F" to "Figure 1F".

P16, 8 lines from the bottom. Please add "the" between "re-enter" and "cell".

What happened to the cilium in Fig 2B, 0h? Did you often see these split cilia?

P16, last line and first lines of P17. Please check the fig 2 lettering. Fig 2E and F should be C and D.

P17, line 11: please explain "as previously".

P21, line 4, please remove 1x "that".

Reviewer #2 (Remarks to the Author):

The authors present persuasive evidence in this study for the role of the second messenger PIP2 in the ciliary membrane regulating the length and fission of the primary cilium, identifying and confirming several downstream effectors which mediate these responses. The authors used a genetic approach to tease out the components involved, and experimental perturbations with inhibitors, over-expression and inactive mutants, and gene silencing, provide substantial evidence towards their final models of the pathways regulating ciliary length and fission. A consistent detraction from this manuscript is the lack of consideration of recent literature on pathways regulating primary cilium length. The Introduction and Discussion should be updated accordingly. The final model should incorporate advances of others.

Major criticisms:

- The authors did not consider regulated microtubule dynamics in the mechanism for mediating primary cilium length in their final model. It is very clear from the work of Janke and others that tubulin glycylation and glutamylation are critical to this process. The model in Fig. 7B is far too simplistic. It does not incorporate the microtubule cytoskeleton in any way. The authors did cite a relevant paper or two but did not incorporate those discoveries in any serious way in the manuscript. How could the PIP2 signaling pathway regulate tubulin tyrosine ligase-like enzymes? Repeating the PIP2 synthesis and depletion experiments with the new glycylation antibody from Gadadhar (28687664) is important and would greatly improve the impact of this manuscript.
- Other proteins reported as important for cilia length regulation include TSC1 & TSC2 (Rosengren T., et al., 2018); CDC14A (Uddin B., et al., 2018); RCAN-2 (Stevenson N.L., et al., 2018); Gli2 (Hsiao, C.-J., et al., 2018); SCA11 (Bowie, E., et al., 2018); MCHR1 (Hamamoto, A., et al., 2018); FHDC1 (Copeland S.J., et al., 2018). These advances simply must be incorporated.
- There is an over-reliance of inhibitors to confirm aurora kinase A and HDAC function downstream of

PIP2 instead of using siRNA or Crispr.

- Direct measurement of protein-protein interactions would be more persuasive evidence for this pathway, perhaps proximity tagging with APEX or BIO-ID and proteomics could be useful tools, or siRNA and phosphoproteomics and confirmation of phosphorylated substrates. While this may be beyond the scope of the manuscript, the significance of the work that justifies publication in this journal would be bolstered by such experiments.

Minor criticisms:

- The statement in the Abstract: "...including the observation that the ciliary membrane is continuous with the plasma membrane but differs in its phospholipid composition" is simply incorrect. The Chih B. et al. 2011 paper on the transition zone protecting the cilium as a privileged membrane domain is evidence that the ciliary membrane is not continuous with the plasma membrane (PMID: 22179047). Two recent papers show that the ciliary membrane has a distinct lipid composition from that of the cell body (26276100, 19240119). It is clear by now that the ciliary membrane is a privileged domain.
- For all histograms, consider plotting all individual data points with geometric mean and SD instead, this would show the reader the spread of the data for ciliary fission (% of cells) and vesicle budding (% of cells).
- Some inconsistency in figure labeling e.g. Sup. Fig S2 should have bar above image from 0.5 -2 hr and "+rapa". Again for Sup Fig S5 A and B there is no bar above 3 hr "+ rapa".
- Also for this figure a point mutation in Inp54p for a figure S2C, similar to PIPK (KD) in S1C.
- Sup. S4. Compare A and B and the merge for B is only red and blue whereas merge for A is all three colors. Similarly for Fig S5. Compare A 0h and 3h, and merge for 0 hr is not a merge.

In summary, this manuscript is of potentially high interest not only to primary cilium researchers, but more widely to the cell biology community, as the authors partially elucidate the mechanism of the signaling pathway downstream of PIP2 on cilium length and stability. The authors present a thorough and well-reasoned study, but have not considered the microtubule axoneme, the major cytoskeletal structure, in their final model, and need to update their discussion to include the most recent literature for a more complete understanding of primary cilium dynamics.

Editing errors to address:

- P16-17. Figure 2 referencing is incorrect in text. P16. "(Figure 2A vs. B, quantified in E and F)" should be "(Figure 2A vs. B, quantified in C and D)." Similarly, "(Figure 2A vs. C, quantified in G and H)" should be "(Figure 2A vs. E, quantified in G and H)". Finally, "(Figure 2A vs. D, quantified in I and J)" should be "(Figure 2A vs. F, quantified in I and J)".
- P36. Figure 6 legend. "...incubated with tubacin (E 20uM final) respectively alisertib (F, 200 nM final) for 24 h." The drugs are switched in legend vs. figures E. alisertib and F. tubacin.
- Spelling mistakes:
- P2. Abstract "Using live cell microscopy..." suggest "live"
- P25. Line 2 "...proximal cilium is of similar importance,"
- P40. Supplemental Figure S6: "A region was drawn around the area of the cilium..."
- Figure 1. "Inducible" not "Inducible"

Reviewer #3 (Remarks to the Author):

Stilling et al. make use of conditional PIP2 synthesis and depletion enzymes at the primary cilium to define how this phospholipid influences axoneme length in various functional processes and mutational states. The creation of the conditional tools alone will be of great interest to the field to study rapid changes in axoneme length. Using these tools, the authors find that PIP2 gain promoted ciliary fission and a reduction in axoneme length whereas PIP2 loss promoted axoneme length. PIP2-dependent

length changes require actin polymerization, vesicle exchange with the Golgi, and presumably acetylated tubulin. This is a thorough examination of PIP2's involvement in the structure of the cilium and certainly gives a greater appreciation for phospholipids in the process of ciliary decapitation/fission/agonist-induced vesicle budding, which are likely variants of the same process. Some weaknesses include the PIP2 sensor PLCd1 serving to suppress the PIP2-related phenotypes (making the simultaneous measurement of PIP2 presence and altered phenotypes impossible), the inconsistency of PIPK suppressing axoneme length from experiment-to-experiment, and the absence of functional signaling measurements of the various manipulations. Despite this, the data are robust and would be of great interest to the readership of Communications Biology after a few alterations addressed below.

Major criticisms:

1) The control cells for Fig. 6 do not respond to the various treatments as expected. Addition of Inp54p, TMSb, and alisertib should increase axoneme length in the control experiments as shown throughout the manuscript. The absence of response casts doubt on the effectiveness of the manipulations and their ability to rescue Inpp5e knockdown.

2) Connecting the changes in axoneme length to signaling changes (such as Hedgehog signaling) would be of great value to determine whether suppressing PIP2-dependent axoneme changes or rescuing Inpp5e knockdown is cosmetic in nature or would completely rescue cilia function. However, this may be outside the scope of the manuscript.

Minor criticisms:

1) The length of cilia from NIH3T3 control cells in Fig. 1 and 2 seem abnormally large at approximately 10um. Is the scale bar correct? The length of cilia in Fig. 3 controls seem more appropriate at 3-5um.

2) A size for the scale bar is missing in Fig. 5.

3) Rac1/Cdc42 placement above the back arrow in Fig. 7B should be changed as it implies these Rho GTPases promote actin disassembly.

4) Some minor mistakes are scattered throughout the manuscript, such as: consistency of mCherry capitalization in Fig. 5 and 6, spelling of inducible in Fig. 1 graphic, spelling of fission on pg. 17, spelling of Inpp5e and dependent on pg. 3, etc.

PIP₂ determines length and stability of primary cilia

Reviewers' comments:

Reviewer #1 (Remarks to the Author):

In this manuscript, the authors use the FKBP-FRB system to rapidly change the levels of PIP₂ in the cilium, by inducing the localization of either a PIP₂ kinase or a phosphatase into the cilium. Using live imaging and fixed cells they show that increased levels of PIPK and thus increased levels of PIP₂ in cilia results in reduced ciliary length and increased ciliary fission. In contrast, depleting PIP₂, by recruiting INP54P to the cilium, resulted in longer cilia. The authors identify several factors that mediate the effects of increased PIP₂ on cilium length including actin polymerization, most likely in the cilium.

Next, the authors compare their results to a previously identified pathway that regulates shedding of vesicles containing ligand activated receptors when retrograde transport is blocked. Their results suggest that these are indeed very similar processes. Finally, the authors extrapolate their results to Joubert syndrome, a ciliopathy that can be caused by loss of function of INPP5e.

The manuscript reports very interesting results and in principle contributes an important step forward in our understanding of the regulation of cilium length, and links this to a ciliopathy and cellular signaling. However, some of the data presented in the manuscript in its current form are not sufficiently convincing.

-> *Thank you.*

Major points

Although figure 1 nicely shows recruitment of PIPK or INP54P to the cilium, I'm not convinced by the results presented in Sup fig 1 and 2. Especially in Sup Fig 1 I don't see much change in the length of the blue region. The authors have quantified these effects in S1b and S2b. Are these changes significant?

-> *We revised supplemental Figures S1 and S2 in order to better show the extension of the PH domain in the cilium. We analyzed the extension of the PH domain in absolute values (μm , Figures S1B/C, S2B) and relative to the length of the cilium (Figures S1D and S2C). For recruitment of PIPK, the change in the absolute length of the PH domain is significant with recruitment of WT PIPK (Figure S1B, $p=0.0188$, one-way ANOVA) but not for KD PIPK (Figure S1C, $p=0.7648$, one-way ANOVA). The change in relative length of the PH domain was also significant (Figure S2C, $p=0.0313$, one-way ANOVA). For recruitment of the phosphatase *Inp54p*, the change in relative length of the PH domain was significant (Figure S2C, $p=0.0313$, one-way ANOVA), but the absolute length of the PH domain was not quite significant (Figure S2B, $p=0.06487$, one-way ANOVA), possibly due to the small values. The text was adapted on page 14.*

The authors define a "fission" event by the appearance of at least one vesicle in the region of the ciliary tip, but also by a decrease in cilium length by more than 20% of its initial length. Although this has been clearly stated in M&M and results, I think the authors should be more careful. Is a decrease in more than 20% really fission; why not rapid resorption? Would it be possible to show this? In addition, the authors should show what fraction of their data consists of "real" fission events and what fraction are the rapid length reductions.

-> *We have reanalyzed the data and assessed separately the frequency of a length reduction by more than 20% and the appearance of a vesicle at the ciliary tip. We have therefore replaced Figures 1E and 4E. The new panels show the frequency of vesicle appearance (black bars) stacked with the frequency of length reduction (grey bars).*

With PIPK recruitment (Figure 1E), vesicle appearance is much more common than length reduction ($p < 0.0001$, two-way ANOVA). Both events are more common in WT PIPK than in KD PIPK ($p < 0.0001$, two-way ANOVA), but the difference results mainly from differences in vesicle appearance ($p < 0.0001$, two-way ANOVA). The same pattern was observed with serotonin application (Figure 4E). We also adapted the text on page 15/16 and 21.

The authors quantify the length of the cilia in Figure 1F. Please indicate if these changes in length are significant. I think that including the length measurements of the cilia that underwent fission is not useful, as this includes the cilia that were reduced in length by more than 20%.

-> We have modified Figure 1F to show (i) the overall length change and (ii) the length change in those cilia that show occurrence of a vesicle. The latter is interesting because WT PIPK mainly induced the appearance of a vesicle and we were interested to see whether this process was associated with a length change, i.e. whether the length of the cilium was reduced by vesicle secretion or not. This was the case, the length of cilia with vesicle secretion decreased by about 40%.

In Figure 2 important controls are missing. What happens when cells are incubated only with LatA, RAC1(DN), CDC42(DN) and ciliary TMS β ? Similarly, the authors should show the results of incubations with only alisertib or tubacin without addition of rapamycin in Figure 3, and only with BFA in Figure S8. And the same applies to results presented in Figure 4

-> We have added the requested control experiments for cells incubated with LatA or expressing Rac1(DN), CDC42(DN) or ciliary TMS β but without recruitment of PIPK (PIPK and membrane anchor were expressed but rapamycin was not added). To preserve clarity in Figure 2, we added these experiments to supplemental Figure S4 (panels D-L). Similarly, we included controls for alisertib or tubacin only in supplemental Figure S4 (panels M-Q). The findings are described on pages 18-19 in the main text.

In Figure 4, serotonin was added in presence of DN dynamin to block endocytosis. We now added new panels 4G and 4H to describe the effects of serotonin with WT instead of DN dynamin: Serotonin does not induce vesicle budding and does not change ciliary length. As for the other manipulations, serotonin was controlled by buffer only application in the same panels 4E and 4F. The effects of TMS β are described in the new supplemental Figure S4G, K, L mentioned above. The effects of Inp54p are described in detail in Figure 1D and 1G. The findings are discussed on page 22 in the main text.

The authors conclude that the effect of PIP₂ on cilium length requires actin polymerization, importantly in the cilium. Very interesting, but to me a bit preliminary. Would it be possible to prove this in an independent way, or e.g. show actin polymerization in the cilium?

-> The main focus of this manuscript is ciliary PIP₂. Still, we do think that the case for a role of actin in PIP₂-mediated length changes is pretty strong. (1) There is ample literature for a role of actin in ciliary length changes and for regulation of actin by PIP₂. (2) Pharmacological blockade of actin polymerization blocked PIP₂-induced length changes. (3) Expression of TMS β , a protein that blocks actin polymerization locally in the cilium, blocked PIP₂-induced length changes. (4) In a previous study, Phua et al. 2017, we visualized actin polymerization after serum addition using the lifeact probe. Actin polymerization was identified as a fast and transient event. We therefore did not expect to observe actin polymerization with the low temporal resolution we used for this study.

On page 19 the authors state that they observed budding events in 48.15 + 15.65% of cells exposed to serotonin but in less than 10% of the cells incubated with buffer. However, these results are not significant, as indicated in Figure 4. This should also be stated in the text.

PIP₂ determines length and stability of primary cilia

-> *We increased the number of independent experiments included in Figure 4. The differences in Figure 4E are now statistically significant (two-way ANOVA).*

At the beginning of page 19 the authors conclude that agonist induced vesicle budding (which is not significantly different from non-agonist induced “budding”) can be induced by blocking endocytosis. However, this has not been tested: there are no data of cells treated with vs without dynamin (DN) to show that this affects vesicle budding.

-> *We added additional data demonstrating that agonist application does not induce vesicle budding in cells expressing WT dynamin. The new data was included into Figure 4 (panels G and H). The text was adapted on page 22.*

In addition, it is important that the authors reproduce the published results, by using latrunculin A, to show if latrunculin A affects vesicle budding to the same extent as DN-dynamin, and also to combine the two to see if they have additive effects.

-> *We used TMS β to demonstrate that actin polymerization in the cilium is necessary for agonist-induced vesicle budding (Figure 4G and 4H). We think that this is a more specific manipulation than exposure to latrunculin A. TMS β was also used in combination with DN dynamin (Figure 4I, 4J), but we note that they have opposite effects: DN dynamin is required for budding whereas TMS β blocks budding.*

Page 21, Figure 6G-J. These figures indicate there are significant differences, but it is not clear which results are significantly different, and to what extent. The ANOVA shows there are significant differences, but this should be followed by a post-hoc analysis that reveals the significance of specific comparisons.

-> *We have now included the results of the posthoc tests in the figure and in the legend. Figure 6G, H: In control conditions, Inpp5e siRNA cilia are significantly **shorter** than control siRNA cilia ($p < 0.0001$) whereas with TMS β or Inp54p Inpp5e siRNA cilia are significantly **longer** than control siRNA cilia ($p < 0.0001$).*

*Figure 6 I, J: In control conditions, Inpp5e siRNA cilia **are** significantly shorter than control siRNA cilia ($p < 0.0001$), with alisertib, Inpp5e siRNA cilia **are not** significantly shorter than control siRNA cilia. With tubacin, Inpp5e siRNA cilia **are still** significantly shorter than control siRNA cilia ($p = 0.0032$) in the posthoc test, but the significant interaction between the factors treatment and siRNA ($p < 0.0001$) demonstrates that tubacin still alters the Inpp5e siRNA phenotype. The interaction remains with the same p value even if one compares only tubacin and control (excluding alisertib).*

Minor points

P4, end of 2nd paragraph: please change GPCR for GPCRs

P4, end of 3rd paragraph: please add “the” between “of” and “general”

P5, line 3. The authors state that in the proximal ciliary membrane the amount of PIP₂ is “normal”. “normal” is not the right term. The authors should better use “as in the plasma membrane” or something similar. The same applies to line 5 of the abstract.

P7, plasmids, 2nd paragraph. Please correct “PIPK (which phosphorylates PIP₂ to PIP)”.

P14, line 14, please change “Figure 2F” to “Figure 1F”.

P16, 8 lines from the bottom. Please add “the” between “re-enter” and “cell”.

What happened to the cilium in Fig 2B, 0h? Did you often see these split cilia?

P16, last line and first lines of P17. Please check the fig 2 lettering. Fig 2E and F should be C and D.

P17, line 11: please explain “as previously”.

P21, line 4, please remove 1x “that”.

-> *Thank you for these comments. They were corrected.*

With respect to the split cilium in Figure 2F, these were very rare. We therefore replaced this panel.

Reviewer #2 (Remarks to the Author):

The authors present persuasive evidence in this study for the role of the second messenger PIP₂ in the ciliary membrane regulating the length and fission of the primary cilium, identifying and confirming several downstream effectors which mediate these responses. The authors used a genetic approach to tease out the components involved, and experimental perturbations with inhibitors, over-expression and inactive mutants, and gene silencing, provide substantial evidence towards their final models of the pathways regulating ciliary length and fission. A consistent detractor from this manuscript is the lack of consideration of recent literature on pathways regulating primary cilium length. The Introduction and Discussion should be updated accordingly. The final model should incorporate advances of others.

-> Thank you for the comments. We revised the introduction as suggested.

For the cartoons in Figure 7, as indicated by the Figure title, our aim was to provide a graphic summary of this manuscript. We agree that a model should include many more aspects. Providing a full model of ciliary fission is beyond the scope of this manuscript. Still, we included microtubules and microtubule regulation in panel 7A and changed the title of panel 7B from "Membrane model of ciliary fission" to "Proposed mechanism of ciliary membrane fission".

Moreover, we toned down some statements that could be misleading: In the last paragraph we wrote that PIP₂ is "a unifying point" instead of "the unifying point", thus using the same word as in the last paragraph of the introduction ("a central regulator").

Major criticisms:

- The authors did not consider regulated microtubule dynamics in the mechanism for mediating primary cilium length in their final model. It is very clear from the work of Janke and others that tubulin glycylation and glutamylation are critical to this process. The model in Fig. 7B is far too simplistic. It does not incorporate the microtubule cytoskeleton in any way. The authors did cite a relevant paper or two but did not incorporate those discoveries in any serious way in the manuscript. How could the PIP₂ signaling pathway regulate tubulin tyrosine ligase-like enzymes? Repeating the PIP₂ synthesis and depletion experiments with the new glycylation antibody from Gadadhar (28687664) is important and would greatly improve the impact of this manuscript.
- Other proteins reported as important for cilia length regulation include TSC1 & TSC2 (Rosengren T., et al., 2018); CDC14A (Uddin B., et al., 2018); RCAN-2 (Stevenson N.L., et al., 2018); Gli2 (Hsiao, C-J., et al., 2018); SCA11 (Bowie, E., et al., 2018); MCHR1 (Hamamoto, A., et al., 2018); FHDC1 (Copeland S.J., et al., 2018). These advances simply must be incorporated.

-> Reductionist approaches have been very successful and for this reason we focused on the role of the ciliary membrane for this manuscript - just as the cited work focused on the cytoskeleton and other proteins. As we have already noted in the original discussion in reference to Figure S6, the length of ciliary membrane and microtubules must be tightly correlated, and it will be interesting to observe in subsequent studies what mediates this co-regulation. We currently have too little data about this mechanism, but we believe that the manipulation we established to induce growth of ciliary membrane followed by growth of the microtubules provides an exciting paradigm to study this question. We had already speculated that HDAC6, which regulates posttranslational tubulin modification, could be one

of the molecular links between our pathway and the tubulin code. Still we agree that recent literature should be cited. The work by Janke was already cited (Gadadhar 2016). The work by Bowie refers to the tau tubulin kinase SCA11 and is now also cited in the context of microtubule regulation (page 30).

The work by Uddin (CDC14A) and Copeland (FHDC1) describe regulators of actin and are now cited in this context (page 4 and page 29), and we note that the actin cytoskeleton was already a focus of this manuscript.

We thank the reviewer for pointing out the work by Hamamoto, which demonstrates that ciliary shortening (and thus potentially vesicle budding) can be initiated by some agonists even without blockade of endocytosis or retrograde transport (MCHR1 is a GPCR for melanin-concentrating hormone). It is tempting to speculate that the molecular mechanism might be similar as for the serotonin receptor, but the evidence is too preliminary to be concluded in this manuscript and the work beyond the scope of this revision. We note, however, that the pathway proposed by Hamamoto involved Akt, which is activated by the membrane lipid PIP₃, which is made from PIP₂, and we have some preliminary evidence that the PI 3-kinase could be involved in vesicle budding. This will certainly be interesting to determine in further studies. We cite the paper on page 5 and discuss this possibility on page 22.

- There is an over-reliance of inhibitors to confirm aurora kinase A and HDAC function downstream of PIP₂ instead of using siRNA or Crispr.

-> We have used genetic manipulations wherever possible, but constitutively knocking out or knocking down aurora kinase A or HDAC in a dividing cell line will produce many off-target effects and compensatory changes. The advantages of the small molecules we used is that they could be applied acutely, thus avoiding compensatory changes. We also note that they are established agents for this purpose. Furthermore, aurora kinase and HDAC are not the main focus of this manuscript, and that we interpreted the findings carefully.

- Direct measurement of protein-protein interactions would be more persuasive evidence for this pathway, perhaps proximity tagging with APEX or BIO-ID and proteomics could be useful tools, or siRNA and phosphoproteomics and confirmation of phosphorylated substrates. While this may be beyond the scope of the manuscript, the significance of the work that justifies publication in this journal would be bolstered by such experiments.

-> We do not see how protein-protein interaction or proteomic studies can advance the main point of this article that the membrane lipid PIP₂ is a key regulator of ciliary length. (1) PIP₂ is not a protein. (2) Protein-protein interactions or proteomics cannot prove that a signaling step is necessary for an observed effect.

Minor criticisms:

- The statement in the Abstract: "...including the observation that the ciliary membrane is continuous with the plasma membrane but differs in its phospholipid composition" is simply incorrect. The Chih B. et al. 2011 paper on the transition zone protecting the cilium as a privileged membrane domain is evidence that the ciliary membrane is not continuous with the plasma membrane (PMID: 22179047). Two recent papers show that the ciliary membrane has a distinct lipid composition from that of the cell body (26276100, 19240119). It is clear by now that the ciliary membrane is a privileged domain.

-> We agree that the ciliary membrane is a privileged domain and has a distinct lipid composition. In fact, the focus of this article is to determine the functional significance of one of these differences. Still, in EM images the ciliary membrane is continuous with the plasma membrane, and it is not entirely resolved what underlies the differences in lipid and protein composition. The cited references are of course relevant, but the issue is far from being resolved.

(1) *It is well established that membrane proteins such as smoothend diffuse from the plasma membrane into the cilium (Milenkovic et al., 2009 and 2015; doi 10.1083/jcb.200907126 and 10.1073/pnas.1510094112; Takao et al., 2014, doi 10.1016/j.cub.2014.08.012). In fact, most ciliary membrane proteins enter the cilium by exocytosis in the ciliary pocket, which is a part of the plasma membrane, followed by lateral transport through the transition zone to the ciliary membrane (e.g. reviews by Emmer 2010, Garcia-Gonzalo 2012, Hsiao 2012). This mechanism requires a continuous membrane.*

(2) *If there was a tight fence between plasma membrane and ciliary membrane, localized enzymes such as Inpp5e would not be required. Their presence indicates that there is a constant inflow of PIP₂ that needs to be dephosphorylated.*

(3) *There might in addition be species differences as we demonstrated that the confinement of PIP₂ to the transition zone published for drosophila cilia was not observed in our system (Figure S3).*

- For all histograms, consider plotting all individual data points with geometric mean and SD instead, this would show the reader the spread of the data for ciliary fission (% of cells) and vesicle budding (% of cells).

-> *We followed the Nature standards and plotted individual values.*

- Some inconsistency in figure labeling e.g. Sup. Fig S2 should have bar above image from 0.5 -2 hr and "+rapa". Again for Sup Fig S5 A and B there is no bar above 3 hr "+ rapa".
- Also for this figure a point mutation in Inp54p for a figure S2C, similar to PIPK (KD) in S1C.
- Sup. S4. Compare A and B and the merge for B is only red and blue whereas merge for A is all three colors. Similarly for Fig S5. Compare A 0h and 3h, and merge for 0 hr is not a merge.

-> *The bars were added.*

Figure S5 investigates the effect of the PH(PLCd1) domain on ciliary fission. There is no established point mutation in PH(PLCd1) that disrupts PIP₂ binding.

Figure S4 compares ciliary length by 5HT6 and acetylated tubulin. We therefore included only these channels in both merged images. In Figure S5A, color mixing was adjusted to show that it is a merged image.

In summary, this manuscript is of potentially high interest not only to primary cilium researchers, but more widely to the cell biology community, as the authors partially elucidate the mechanism of the signaling pathway downstream of PIP₂ on cilium length and stability. The authors present a thorough and well-reasoned study, but have not considered the microtubule axoneme, the major cytoskeletal structure, in their final model, and need to update their discussion to include the most recent literature for a more complete understanding of primary cilium dynamics.

-> *Thank you for the comment. As noted above we have expanded introduction and discussion to include the literature on microtubules.*

Editing errors to address:

- P16-17. Figure 2 referencing is incorrect in text. P16. "(Figure 2A vs. B, quantified in E and F)" should be "(Figure 2A vs. B, quantified in C and D)." Similarly, "(Figure 2A vs. C, quantified in G and H)" should be "(Figure 2A vs. E, quantified in G and H). Finally, "(Figure 2A vs. D, quantified in I and J)" should be "(Figure 2A vs. F, quantified in I and J)".
- P36. Figure 6 legend. "...incubated with tubacin (E 20uM final) respectively alisertib (F, 200 nM final) for 24 h." The drugs are switched in legend vs. figures E. alisertib and F. tubacin.
- Spelling mistakes:
- P2. Abstract "Using life cell microscopy..." suggest "live"
- P25. Line 2 "...proximal cilium is of similar importance,"

PIP₂ determines length and stability of primary cilia

- P40. Supplemental Figure S6: “A region was drawn around the area of the cilium...”
- Figure 1. “Inducible” not “Inducible”

-> Corrected

Reviewer #3 (Remarks to the Author):

Stilling et al. make use of conditional PIP₂ synthesis and depletion enzymes at the primary cilium to define how this phospholipid influences axoneme length in various functional processes and mutational states. The creation of the conditional tools alone will be of great interest to the field to study rapid changes in axoneme length. Using these tools, the authors find that PIP₂ gain promoted ciliary fission and a reduction in axoneme length whereas PIP₂ loss promoted axoneme length. PIP₂-dependent length changes require actin polymerization, vesicle exchange with the Golgi, and presumably acetylated tubulin. This is a thorough examination of PIP₂'s involvement in the structure of the cilium and certainly gives a greater appreciation for phospholipids in the process of ciliary decapitation/fission/agonist-induced vesicle budding, which are likely variants of the same process. Some weaknesses include the PIP₂ sensor PLCd1 serving to suppress the PIP₂-related phenotypes (making the simultaneous measurement of PIP₂ presence and altered phenotypes impossible), the inconsistency of PIPK suppressing axoneme length from experiment-to-experiment, and the absence of functional signaling measurements of the various manipulations. Despite this, the data are robust and would be of great interest to the readership of *Communications Biology* after a few alterations addressed below.

Major criticisms:

1) The control cells for Fig. 6 do not respond to the various treatments as expected. Addition of Inp54p, TMSb, and alisertib should increase axoneme length in the control experiments as shown throughout the manuscript. The absence of response casts doubt on the effectiveness of the manipulations and their ability to rescue Inpp5e knockdown.

-> Blocking components of the pathway we proposed, i.e. Inp54p, TMSb, alisertib, tubacin always blocked ciliary shortening induced by PIP₂ synthesis, by agonist, and by Inpp5e knockdown. In contrast, the effects of Inp54p, TMSb, alisertib, tubacin in the absence of further manipulations are somewhat heterogenous: Inp54p alone induced ciliary elongation in Figure 1D/G but not in Figure 6G, but Inp54p recruitment was used for Figure 1 whereas Figure 6G is with constitutive 5HT6-Inp54p expression. TMSb alone showed a nonsignificant trend for ciliary elongation in Figure S4F, but not in Figure 6G. Alisertib alone showed a significant ciliary elongation in Figure S4N, but not in Figure 6I. Tubacin showed a significant ciliary elongation in Figure S4O and also in Figure 6I.

A consistent increase in ciliary length by Inp54p, TMSb, alisertib, tubacin would be a strong argument for the existence of basal or constitutive ciliary fission occurring by the same pathway. Because our findings were inconsistent, our interpretation in this regard was careful, and we have revised and somewhat expanded this section (page 27), explicitly pointing out the remaining inconsistencies on page 28.

Yet, we disagree that this inconsistency puts the entire dataset into question. Basal or constitutive ciliary fission is a rare event, occurring in about 10% of cilia over 3 h (Figure 1E). Its extent is likely affected by small experimental differences such as cell passage, and it is the extent of basal fission that determines the effects of Inp54p, TMSb, alisertib, tubacin in the absence of further manipulations.

2) Connecting the changes in axoneme length to signaling changes (such as Hedgehog signaling) would be of great value to determine whether suppressing PIP₂-dependent axoneme changes or rescuing Inpp5e knockdown is cosmetic in nature or would completely

PIP₂ determines length and stability of primary cilia

rescue cilia function. However, this may be outside the scope of the manuscript.

-> This is an interesting suggestion that we will certainly follow in the future. Yet, we agree that it is beyond the scope of the current manuscript.

Minor criticisms:

1) The length of cilia from NIH3T3 control cells in Fig. 1 and 2 seem abnormally large at approximately 10µm. Is the scale bar correct? The length of cilia in Fig. 3 controls seem more appropriate at 3-5µm.

-> The scale bar is correct, cilia in these experiments were quite long.

2) A size for the scale bar is missing in Fig. 5.

-> This was corrected.

3) Rac1/Cdc42 placement above the back arrow in Fig. 7B should be changed as it implies these Rho GTPases promote actin disassembly.

-> This is indeed misleading. We had placed Rac1/Cdc42 at this location because it was shown to promote PIP₂ synthesis, the "circular" arrows were intended to indicate a positive feedback loop. We replaced it by a single arrow to avoid this problem.

4) Some minor mistakes are scattered throughout the manuscript, such as: consistency of mCherry capitalization in Fig. 5 and 6, spelling of inducible in Fig. 1 graphic, spelling of fission on pg. 17, spelling of Inpp5e and dependent on pg. 3, etc.

-> The manuscript was carefully revised for spelling and the noted issues resolved.

Reviewers' comments:

Reviewer #1 (Remarks to the Author):

In this revised manuscript the authors have dealt with most of my comments.

One remaining concern are the control experiments for the data presented in figures 2, 3 and 4, now shown in figure S4. I agree that adding the pictures to figures 2, 3 and 4 would complicate these figures too much. But the quantifications of the control experiments, shown in figure S4 H, I, J, K, L should be added to Figure 2, those shown in figure S4O, P, Q to figure 3D-G, and the TMS β control should also be included in figure 4I and J.

In addition, I am lost with the response to my comment where the authors state:

"-> We used TMS β to demonstrate that actin polymerization in the cilium is necessary for agonist-induced vesicle budding (Figure 4G and 4H). We think that this is a more specific manipulation than exposure to latrunculin A. TMS β was also used in combination with DN dynamin (Figure 4I, 4J)"
Figure 4G and H show the effects of treatment with dynamin (WT) and serotonin, not TMS β .
I don't see the results where they combine DN dynamin combined with TMS β , but perhaps this is because DN dynamin is not indicated in figure 4I and J? Please clarify this.

Please correct the sentence "When TMS β was expressed in cilia but PIPK to recruited, we observed" on page 17.

Reviewer #2 (Remarks to the Author):

I found the responses to reviewers to be more than adequate and the revised manuscript much improved.

Reviewer #3 (Remarks to the Author):

The revised manuscript by Stilling et al. shows marked improvement to many of their claims that phospholipids influence axoneme length. Many controls have been added as detailed in their rebuttal letter that improve the confidence in the underlying data. However, I am still unconvinced by the discrepancy in the control experiments between various figures and the response by the authors to the criticism.

The claim that their interpretation of the data was "careful" because they noted the inconsistency does not mean that the initial experiments were not flawed, i.e. the cells were not treated in the same way or the drugs were degraded/less potent over time. To have the treatments always work in certain experiments when a factor is manipulated but be heterogenous in the absence of manipulations is quite confusing. This criticism is not only restricted to the Inp54p, TMSb, and alisertib in Figure 6, but extends to the differences in cilia length from NIH3T3 control cells in Figures 1-3, where different cells or distinct culture conditions/treatments would explain the data better than stochastic differences in cilia length that happen to have similar standard deviations. These discrepancies are not addressable by text changes alone that state "inconsistencies remain to be solved in future studies" and need to be carefully addressed by repeating the experiments.

*To submit your data to Dryad, please use the link below, which will take you directly to your entry in

the Dryad repository. Using this link will ensure that reviewers will have access to your data during the review process:

<http://datadryad.org/submit?journalID=COMMSBIO&manu=COMMSBIO-18-1478A>

Once your data package is deposited, a unique and stable identifier (DOI) will be sent to you and to the journal for inclusion in the published article. For questions or feedback on the Dryad data submission process, please email help@datadryad.org.

Reviewer 1

Reviewer 1: One remaining concern are the control experiments for the data presented in figures 2, 3 and 4, now shown in figure S4. I agree that adding the pictures to figures 2, 3 and 4 would complicate these figures too much. But the quantifications of the control experiments, shown in figure S4 H, I, J, K, L should be added to Figure 2, those shown in figure S4O, P, Q to figure 3D-G, and the TMS β control should also be included in figure 4I and J.

Response: We added the panels to the main figures as suggested and revised the statistical analysis to match the new display (i.e. ANOVA instead of t-test).

Reviewer 1: In addition, I am lost with the response to my comment where the authors state: "-> We used TMS β to demonstrate that actin polymerization in the cilium is necessary for agonist-induced vesicle budding (Figure 4G and 4H). We think that this is a more specific manipulation than exposure to latrunculin A. TMS β was also used in combination with DN dynamin (Figure 4I, 4J)"

Figure 4G and H show the effects of treatment with dynamin (WT) and serotonin, not TMS β . I don't see the results where they combine DN dynamin combined with TMS β , but perhaps this is because DN dynamin is not indicated in figure 4I and J? Please clarify this.

Response: The combination of DN dynamin with TMS β is depicted in Figures 4C, 4I and 4J. We apologize for the confusion in the labels, which resulted from the fact that we introduced the two panels G and H with WT dynamin into Figure 4 during the last revision.

As depicted in Figure 4A-D and described in the Figure legend, all panels in Figure 4 include DN dynamin except panels G and H, but we can see how the label "Dynamin (WT)" in panels G and H might obscure that fact that dynamin DN was present in all the other panels.

Reviewer 1: Please correct the sentence "When TMS β was expressed in cilia but PIPK to recruited, we observed" on page 17.

Response: Corrected ("to" -> "not")

Reviewer 3

Reviewer 3:

The revised manuscript by Stilling et al. shows marked improvement to many of their claims that phospholipids influence axoneme length. Many controls have been added as detailed in their rebuttal letter that improve the confidence in the underlying data. However, I am still unconvinced by the discrepancy in the control experiments between various figures and the response by the authors to the criticism.

The claim that their interpretation of the data was "careful" because they noted the inconsistency does not mean that the initial experiments were not flawed, i.e. the cells were not treated in the same way or the drugs were degraded/less potent over time. To have the treatments always work in certain experiments when a factor is manipulated but be heterogeneous in the absence of manipulations is quite confusing. This criticism is not only restricted to the Inp54p, TMSb, and alisertib in Figure 6, but extends to the differences in cilia length from NIH3T3 control cells in Figures 1-3, where different cells or distinct culture conditions/treatments would explain the data better than stochastic differences in cilia length that happen to have similar standard deviations. These discrepancies are not addressable by text changes alone that state "inconsistencies remain to be solved in future studies" and need to be carefully addressed by repeating the experiments.

Response:

In order to rule out that the selection of cilia for the time-lapse experiments introduced a bias into the results depicted in Figures 1-3, we performed new experiments with constitutive expression of 5HT6-PIPK, i.e. a PIP kinase expressed in primary cilia by the ciliary anchor 5HT6. We then determined ciliary length in a large and unbiased sample of several hundred transfected cells using (a) the fluorescence of the ciliary marker 5HT6-YFP and (b) the staining for acetylated tubulin as readouts. In order to show the range of observed ciliary lengths, the results are depicted as violin plots of all cilia analyzed in 3 independent experiments (new Figure 6B, E, I, L). Each violin represents between 175 and 241 cilia. For statistical analysis we determined for each independent experiment the median of all cilia per group and plotted mean +/- SEM of these medians.

With this new approach, we demonstrated that expression of PIPK reduces ciliary length (solid vs. open violins and solid vs open bars). This reduction was alleviated by co-expression of TMSb (Figure 6B-D) and by incubation with Tubacin or Alisertib (Figure 6I-N). The addition of DMSO alone had no effect (Figure 6E-G). These findings confirm our main conclusion from Figure 1-3, that an increase in ciliary PIP₂ reduces ciliary length and that this effect is dependent on ciliary actin, AurkA and HDAC6.

In the new experiments, we did not observe the “overshoot” seen in the original Figure 6 for the groups with Inpp5e siRNA and Inp54p, TMSb or tubacin. For this reason, we moved the original Figure 6 into the supplement as supplemental Figure S9.

REVIEWERS' COMMENTS:

Reviewer #3 (Remarks to the Author):

The authors have satisfied my criticisms and I recommend acceptance.